# Natural Actor-Critic for Robust Reinforcement Learning with Function Approximation

**Ruida Zhou**[*]
Texas A&M University
ruida@tamu.edu

**Tao Liu**[*]
Texas A&M University
tliu@tamu.edu

**Min Cheng**
Texas A&M University
minrara0404@tamu.edu

**Dileep Kalathil**
Texas A&M University
dileep.kalathil@tamu.edu

**P. R. Kumar**
Texas A&M University
prk@tamu.edu

**Chao Tian**
Texas A&M University
chao.tian@tamu.edu

## Abstract

We study robust reinforcement learning (RL) with the goal of determining a well-performing policy that is robust against model mismatch between the training simulator and the testing environment. Previous policy-based robust RL algorithms mainly focus on the tabular setting under uncertainty sets that facilitate robust policy evaluation, but are no longer tractable when the number of states scales up. To this end, we propose two novel uncertainty set formulations, one based on double sampling and the other on an integral probability metric. Both make large-scale robust RL tractable even when one only has access to a simulator. We propose a robust natural actor-critic (RNAC) approach that incorporates the new uncertainty sets and employs function approximation. We provide finite-time convergence guarantees for the proposed RNAC algorithm to the optimal robust policy within the function approximation error. Finally, we demonstrate the robust performance of the policy learned by our proposed RNAC approach in multiple MuJoCo environments and a real-world TurtleBot navigation task.

## 1 Introduction

Training a reinforcement learning (RL) algorithm directly on a real-world system is expensive and potentially risky due to the large number of data samples required to learn a satisfactory policy. To overcome this issue, RL algorithms are typically trained on a simulator. However, in most real-world applications, the nominal model used in the simulator model may not faithfully represent the real-world system due to various factors, such as approximation errors in modeling or variations in real-world parameters over time. For example, the mass, friction, sensor/actuator noise, and floor terrain in a mobile robot simulator may differ from those in the real world. This mismatch, called simulation-to-reality-gap, can significantly degrade the performance of standard RL algorithms when deployed on real-world systems [36, 44, 55, 51]. The framework of robust Markov decision process (RMDP) [19, 40] is used to model this setting where the testing environment is uncertain and comes from an uncertainty set around the nominal model. The optimal robust policy is defined as the one which achieves the optimal worst-case performance over all possible models in the uncertainty set. The goal of robust reinforcement learning is to learn such an optimal robust policy using only the data sampled from the simulator (nominal) model.

The RMDP planning problem has been well studied in the tabular setting [63, 62, 35, 46, 16]. Also, many works have developed model-based robust RL algorithms in the tabular setting [66, 69, 42,

---

[*]The first two authors contributed equally.

37th Conference on Neural Information Processing Systems (NeurIPS 2023).

64, 49], focusing on sample complexity. Many robust Q-learning algorithms [45, 33, 39, 59, 30] and policy gradient methods for robust RL [61, 27, 29, 58, 17] have been developed for the tabular setting . Different from all these works, the main goal of this paper is to develop a computationally tractable robust RL algorithm with provable convergence guarantees for RMDPs with large state spaces, using linear and nonlinear function approximation for robust value and policy.

One of the main challenges of robust RL with a large state space is the design of an effective uncertainty set that is amenable to computationally tractable learning with function approximation. Robust RL algorithms, both in their implementation and technical analysis, require robust Bellman operator evaluations [19, 40] which involve an inner optimization problem over the uncertainty set. Performing this inner optimization problem and/or getting an unbiased estimate of the robust Bellman operator evaluation using only the samples from the nominal model can be intractable for commonly considered uncertainty sets when the state space is large. For example, for $f$-divergence-based uncertainty sets [66, 69, 64, 49, 33, 30], the robust Bellman operator estimate requires solving for a dual variable associated with each state, which is prohibitive for large state spaces. For $R$-contamination [61], the estimate requires calculating the minimum of the value function over state space, which is impractical for large state spaces. It is also infeasible for $\ell_p$ norm-based uncertainty sets [27], where the estimates require calculating the median, mean, or average peak of the value function depending on the choice of $p$. Robust RL with function approximation has been explored in a few works [53, 41, 45, 43, 60, 6]. However, these works explicitly or implicitly assume an oracle that approximately computes the robust Bellman operator estimate, and the uncertainty set design that facilitates computation and learning is largely ignored. We overcome this challenge by introducing two novel uncertainty set formulations, one based on double sampling (DS) and the other on an integral probability metric (IPM). Both are compatible with large-scale robust MDP, with the robust Bellman operators being amenable to unbiased estimation from the data sampled from the nominal model, enabling effective practical robust RL algorithms.

Policy-based RL algorithms [47, 31, 18, 15], which optimize the policy directly, have been extremely successful in learning policies for continuous control tasks. Value-based RL approaches, such as Q-learning, cannot be directly applied to continuous action problems. Moreover, recent advances in policy-based approaches can establish finite time convergence guarantees and also offer insights into the practical implementation [3, 37, 4, 8]. However, most of the existing works on robust RL with function approximation use value-based approaches [53, 41, 45, 43, 60, 34] that are not scalable to continuous control problems. On the other hand, the existing works that use policy-based methods for robust RL with convergence guarantees are limited to the tabular setting [61, 27, 29, 58, 17]. We close this important gap in the literature by developing a novel robust natural actor-critic (RNAC) approach that leverages our newly designed uncertainty sets for scalable learning.

**Summary of Contributions:** $(i)$. We propose two novel uncertainty sets, one using double sampling (Section 3.1) and the other an integral probability metric (Section 3.2), which are both compatible for robust RL with large state space and function approximation. Though robust Bellman operators require the unknown worst-case models, we provide *unbiased* empirical robust Bellman operators that are computationally easy to utilize and only based on samples from the nominal model;

$(ii)$. We propose a novel RNAC algorithm (Section 4) which to the best of our knowledge is the first policy-based approach for robust RL under function approximation, with provable convergence guarantees. We consider both linear and general function approximation for theoretical study, with the latter relegated to Appendix E due to space constraints. Under linear function approximation, the RNAC with a robust critic performing "robust linear-TD" (Section 5) and a robust natural actor performing "robust Q-NPG" (Section 6) is proved to converge to the optimal robust policy within the function approximation error. Specifically, $\tilde{O}(1/\varepsilon^2)$ sample complexity can be achieved, or $\tilde{O}(1/\varepsilon^3)$ for policy update with a constant step size. For the robust linear-TD, we study the contraction behavior of the projected robust Bellman operator for RMDPs with the proposed uncertainty sets and the well-known $f$-divergence uncertainty sets, which we believe is of independent interest;

$(iii)$. We implement the proposed RNAC in multiple MuJoCo environments (Hopper-v3, Walker2d-v3, and HalfCheetah-v3), and demonstrate that RNAC with the proposed uncertainty sets results in robust behavior while canonical policy-based approaches suffer significant performance degradation. We also test RNAC on TurtleBot [5], a real-world mobile robot, performing a navigation task. We show that the TurtleBot with RNAC successfully reaches its destination while canonical non-

robust approaches fail under adversarial perturbations. A video of the demonstration on TurtleBot is available at **[Video Link]** and the RNAC code is provided in the supplementary material.

Due to the page limit, a detailed literature review and comparison of our work with the existing robust RL algorithms are deferred to Appendix G.

## 2  Preliminaries

**Notations:** For any set $\mathcal{X}$, denote by $|\mathcal{X}|$ its cardinality, by $\Delta_\mathcal{X}$ a $(|\mathcal{X}|-1)$-dimensional probability simplex, and by $\text{Unif}(\mathcal{X})$ a uniform distribution over $\mathcal{X}$. Let $[m] := \{1, \ldots, m\}$.

A Markov decision process (MDP) is represented by a tuple $(\mathcal{S}, \mathcal{A}, \kappa, r, \gamma)$, where $\mathcal{S}$ is the state space, $\mathcal{A}$ is the action space, $\kappa = (\kappa_0, \kappa_1, \ldots)$ is a possibly non-stationary transition kernel sequence with $\kappa_t : \mathcal{S} \times \mathcal{A} \to \Delta_\mathcal{S}$, $r : \mathcal{S} \times \mathcal{A} \to [0, 1]$ is the reward function, and $\gamma \in (0, 1)$ is the discount factor. For a stationary policy $\pi : \mathcal{S} \to \Delta_\mathcal{A}$, its value function is $V_\kappa^\pi(s) := \mathbb{E}_{\kappa,\pi} \left[\sum_{t=0}^\infty \gamma^t r(s_t, a_t) | s_0 = s\right]$, where the expectation is taken w.r.t. the trajectory $(s_0, a_0, s_1, a_1, \ldots)$ with $a_t | s_t \sim \pi_{s_t}$ and $s_{t+1} | (s_t, a_t) \sim \kappa_{t, s_t, a_t}$. We can similarly define the state visitation distribution under initial distribution $\rho \in \Delta_\mathcal{S}$, $d_\rho^{\pi,\kappa}(s) := (1 - \gamma)\mathbb{E}_{\kappa,\pi} \left[\sum_{t=0}^\infty \gamma^t \mathbb{1}(s_t = s) | s_0 \sim \rho\right]$; the state-action value function (Q function), $Q_\kappa^\pi(s, a) := \mathbb{E}_{\kappa,\pi} \left[\sum_{t=0}^\infty \gamma^t r(s_t, a_t) | s_0 = s, a_0 = a\right]$, and the advantage function $A_\kappa^\pi(s, a) := Q_\kappa^\pi(s, a) - V_\kappa^\pi(s)$.

**Robust MDP:** A RMDP is represented by a tuple $(\mathcal{S}, \mathcal{A}, \mathcal{P}, r, \gamma)$, where $\mathcal{P}$ is a set of transition kernels known as the *uncertainty set* that captures the perturbations around the nominal stationary kernel $p^\circ : \mathcal{S} \times \mathcal{A} \to \Delta_\mathcal{S}$, and its robust value function is defined as the corresponding worst case:

$$V_\mathcal{P}^\pi(s) := \inf_{\kappa \in \bigotimes_{t \geq 0} \mathcal{P}} V_\kappa^\pi(s). \tag{1}$$

We will assume the following key $(s, a)$-rectangularity condition that is commonly assumed to facilitate dynamic programming ever since the introduction of RMDPs [19, 40]:

**Definition 1.** $\mathcal{P}$ is $(s, a)$-rectangular, if $\mathcal{P} = \bigotimes_{s,a} \mathcal{P}_{s,a}$, for some $\mathcal{P}_{s,a} \subseteq \Delta_\mathcal{S}$.

The corresponding robust Bellman operator $\mathcal{T}_\mathcal{P}^\pi : \mathbb{R}^\mathcal{S} \to \mathbb{R}^\mathcal{S}$ is

$$(\mathcal{T}_\mathcal{P}^\pi V)(s) = \mathbb{E}_{a \sim \pi(\cdot|s)} \left[r(s, a) + \gamma \inf_{p \in \mathcal{P}_{s,a}} p^\top V\right], \quad \forall s \in \mathcal{S}, \ V \in \mathbb{R}^\mathcal{S}, \tag{2}$$

and the Bellman equation for RMDPs is $V = \mathcal{T}_\mathcal{P}^\pi V$, where $V = V^\pi$ is its unique solution from the Banach fixed-point theorem. Typically, $\mathcal{P}_{s,a}$ is taken as a ball $\{\nu \subseteq \Delta_\mathcal{S} : \mathrm{d}(\nu, p_{s,a}^\circ) \leq \delta\}$ around a nominal model $p^\circ$ of the training environment, where $\mathrm{d}(\cdot, \cdot)$ is some divergence measure between probability distributions, and $\delta > 0$ controls the level of robustness.

There is a stationary optimal policy $\pi^*$ that uniformly maximizes the robust value function, i.e., $V_\mathcal{P}^{\pi^*}(s) = \sup\{V_\mathcal{P}^\pi(s) : \text{history dependent } \pi\}, \forall s$ [19, 40]. Thus, without loss of generality, we only need to optimize within stationary policies. Moreover, for any stationary policy $\pi$ there exists a stationary worst-case kernel $\kappa_\pi$ with $V_{\kappa_\pi}^\pi(s) = V_\mathcal{P}^\pi(s), \forall s$. We can define the robust Q-function and robust advantage function as $Q_\mathcal{P}^\pi(s, a) := Q_{\kappa_\pi}^\pi(s, a)$ and $A_\mathcal{P}^\pi(s, a) := A_{\kappa_\pi}^\pi(s, a)$, respectively. When the uncertainty set is clear from the context, we will omit the subscript $\mathcal{P}$ in $V_\mathcal{P}^\pi, Q_\mathcal{P}^\pi$, and $A_\mathcal{P}^\pi$.

To summarize, the motivation in the robust RL framework is to learn the optimal robust policy by only training on a simulator implementing an (unknown) nominal model $p^\circ$ [53, 45, 41], with the robust RL algorithms only having access to data generated from $p^\circ$, and not from any other model in the uncertainty set $\mathcal{P}$.

## 3  Uncertainty Sets for Large State Spaces

Evaluating the robust Bellman operator $\mathcal{T}_\mathcal{P}^\pi$ requires solving the optimization $\inf_{p \in \mathcal{P}_{s,a}} p^\top V$, which is challenging for arbitrary uncertainty sets. Moreover, since we only have access to data generated by the nominal model $p^\circ$, estimating $\mathcal{T}_\mathcal{P}^\pi$ is not straightforward. Previously studied uncertainty sets, such as $R$-contamination, $f$-divergence, and $\ell_p$ norm, are intractable for RMDPs with large state spaces for these reasons (see Appendix B for detailed explanation). We therefore design two uncertainty sets for RMDPs where the robust Bellman operator has a tractable form and can be unbiasedly

estimated by the data sampled from the nominal model, thus making effective learning possible. We will also use function approximation to tractably parameterize policy and value functions.

## 3.1 Double-Sampling (DS) Uncertainty Set

The central difficulty in employing the robust Bellman operator (2) is how to evaluate $\inf_{p \in \mathcal{P}_{s,a}} p^\top V$. Our first method is based on the following key idea for drawing samples that produce an unbiased estimate for it. Let $\{s'_1, s'_2, \ldots, s'_m\}$ be $m$ samples drawn i.i.d. according to the nominal model $p^\circ_{s,a}$. Then, for any given divergence measure $\mathrm{d}(\cdot, \cdot)$ and radius $\delta > 0$, there exists an uncertainty set $\mathcal{P} = \otimes_{s,a} \mathcal{P}_{s,a}$ such that $\inf_{p \in \mathcal{P}_{s,a}} p^\top V = \mathbb{E}_{s'_{1:m} \overset{i.i.d.}{\sim} p^\circ_{s,a}} \left[ \inf_{\alpha \in \Delta_{[m]} : \mathrm{d}(\alpha, \mathrm{Unif}([m])) \leq \delta} \sum_{i=1}^m \alpha_i V(s'_i) \right]$. (See Appendix B.1 for a brief explanation.) We therefore define an empirical robust Bellman operator $\hat{\mathcal{T}}^\pi_{\mathcal{P}}$ corresponding to (2) by

$$(\hat{\mathcal{T}}^\pi_{\mathcal{P}} V)(s, a, s'_{1:m}) := r(s, a) + \gamma \inf_{\alpha \in \Delta_{[m]} : \mathrm{d}(\alpha, \mathrm{Unif}([m])) \leq \delta} \sum_{i=1}^m \alpha_i V(s'_i). \tag{3}$$

Since $(\mathcal{T}^\pi_{\mathcal{P}} V)(s) = \mathbb{E}_{a \sim \pi(\cdot|s), \, s'_{1:m} \overset{i.i.d.}{\sim} p^\circ_{s,a}} \left[ (\hat{\mathcal{T}}^\pi_{\mathcal{P}} V)(s, a, s'_{1:m}) \right]$, $\hat{\mathcal{T}}^\pi_{\mathcal{P}}$ gives an unbiased estimate, which is one key property we use in our robust RL algorithm. We refer to this as "double sampling" since $\sum_{i=1}^m \alpha_i V(s'_i)$ is the expected value when one further chooses one sample $s'$ from $\{s'_1, s'_2, \ldots, s'_m\}$ according to the distribution $\alpha$ on $[m]$ and evaluates $V(s')$. Above, $\alpha$ is a perturbation of $\mathrm{Unif}([m])$ and when $\delta = 0$, $\alpha = \mathrm{Unif}([m])$ and $s' \sim p^\circ_{s,a}$. The uncertainty set $\mathcal{P}$ corresponding to the double sampling is implicitly defined by specifying the choices of $m$, $\mathrm{d}(\cdot, \cdot)$, and $\delta$. Its key advantage is that samples can only be drawn according to the nominal model $p^\circ$.

Double sampling requires sampling multiple next states for a given state-action pair, which can be implemented if the simulator is allowed to set to any state. (All MuJoCo environments [56] support DS.) Since the calculation of $\alpha$ is within $\Delta_{[m]}$, the empirical robust Bellman operator is tractable for moderate values of $m$. We use $m = 2$ in experiments for training efficiency, where for almost all divergences $\mathrm{d}(\cdot, \cdot)$ we can explicitly write

$$(\hat{\mathcal{T}}^\pi_{\mathcal{P}} V)(s, a, s'_{1:2}) = r(s, a) + \gamma \left( 0.5(V(s'_1) + V(s'_2)) - \delta |V(s'_1) - V(s'_2)| \right). \tag{4}$$

Bellman completeness (value function class closed under the Bellman operator) is a key property for efficient reinforcement learning, whereas RMDP with canonical uncertainty sets may violate it. We show in Appendix B.1 that for RMDP with DS uncertainty sets, the linear function approximation class satisfies Bellman completeness if the nominal model is a linear MDP [22].

## 3.2 Integral Probability Metric (IPM) Uncertainty Set

Given some function class $\mathcal{F} \subset \mathbb{R}^{\mathcal{S}}$ including the zero function, the integral probability metric (IPM) is defined by $\mathrm{d}_{\mathcal{F}}(p, q) := \sup_{f \in \mathcal{F}} \{p^\top f - q^\top f\} \geq 0$ [38]. Many metrics such as Kantorovich metric, total variation, etc., are special cases of IPM under different function classes [38].

The robust Bellman operator $\mathcal{T}^\pi_{\mathcal{P}} V$ (2) requires solving the optimization $\inf_{q \in \mathcal{P}_{s,a}} q^\top V$. For an IPM-based uncertainty set $\mathcal{P}$ with $\mathcal{P}_{s,a} = \{q : \mathrm{d}_{\mathcal{F}}(q, p^\circ_{s,a}) \leq \delta\}$, we relax the domain $q \in \Delta_{\mathcal{S}}$ to $\sum_s q(s) = 1$ as done in [28, 27], which incurs no relaxation error for small $\delta$ if $\min_{s'} p^\circ_{s,a}(s') > 0$.

One should choose $\mathcal{F}$ so that it properly encodes information of the MDP and its value functions. We start by considering linear function approximation. Denote by $\Psi \in \mathbb{R}^{\mathcal{S} \times d}$ the feature matrix with rows $\psi^\top(s)$, $\forall s \in \mathcal{S}$. The value function approximation is $V_w = \Psi w \in \mathbb{R}^{\mathcal{S}}$. A good choice of feature vectors encodes information about the state and transition. For example, vectors $\psi(s), \psi(s')$ should be "close" when states $s, s'$ are "similar" and $V^\pi(s), V^\pi(s')$ have small differences. We propose the following function class (with $\ell_2$ as the preferred norm but any other can be used):

$$\mathcal{F} := \{s \mapsto \psi(s)^\top \xi : \xi \in \mathbb{R}^d, \|\xi\| \leq 1\}. \tag{5}$$

Without loss of generality, assume $\Psi$ has full column rank since $d \ll |\mathcal{S}|$, and let the first coordinate of $\psi(s)$ be 1 for any $s$, which corresponds to the bias term of the linear regressor.

**Proposition 1.** *For the IPM with $\mathcal{F}$ in (5), we have $\inf_{q \in \mathcal{P}_{s,a}} q^\top V_w = (p^\circ_{s,a})^\top V_w - \delta \|w_{2:d}\|$.*

---
**Algorithm 1: Robust Natural Actor-Critic**

---
**Input:** $T, \eta^{0:T-1}, K, N$
**Initialize:** $\theta^0$ for policy parameterization and $w_{init}$ for value function approximation
**for** $t = 0, 1, \ldots, T-1$ **do**

    Robust critic updates $w^t$;   //E.g., $w^t$ = RLTD($\pi_{\theta^t}, K$) Algorithm 2
    Robust natural actor updates $\theta^{t+1}$;//E.g., $\theta^{t+1}$ = RQNPG($\theta^t, \eta^t, w^t, N$) Algorithm 3

---

We can thus design the empirical robust Bellman operator, which is an unbiased estimator of $\mathcal{T}_{\mathcal{P}}^{\pi}$ with sample $s'$ drawn from the nominal model:

$$(\hat{\mathcal{T}}_{\mathcal{P}}^{\pi} V_w)(s, a, s') := r(s, a) + \gamma V_w(s') - \gamma\delta\|w_{2:d}\|. \tag{6}$$

Guided by the last regularization term of the empirical robust Bellman operator (6), when considering value function approximation by neural networks we add a similar negative regularization term for all the neural network parameters except for the bias parameter in the last layer.

## 4   Robust Natural Actor-Critic

We propose a robust natural actor-critic (RNAC) approach in Algorithm 1 for the robust RL problem. As its name suggests, there are two components – a robust critic and a robust actor, which update alternately for $T$ steps. At each step $t$, there is a policy $\pi^t$ determined by parameter $\theta^t$. The robust critic updates the value function approximation parameter $w^t$ based on on-policy trajectory data sampled by executing $\pi^t$ on the nominal model with length $K$. The robust actor then updates the policy with step size $\eta^t$, and the critic returns $w^t$ by on-policy trajectory data with length $N$. In practice, a batch of on-policy data can be sampled and used for both critic and actor updates.

We now give the main (informal) convergence results for our RNAC algorithm with linear function approximation and DS or IPM uncertainty sets, where the robust critic performs robust linear-TD and the robust natural actor performs robust-QNPG (as the comments in Algorithm 1). The formal statements, proofs, and generalization to general function approximation are given in Appendix E.

**Theorem 1** (Informal linear convergence of RNAC). *Under linear function approximation, RNAC in Algorithm 1 with DS or IPM uncertainty sets using an RLTD robust critic update and an RQNPG robust natural actor update, with appropriate geometrically increasing step sizes $\eta^t$, achieves $\mathbb{E}[V^{\pi^*}(\rho) - V^{\pi^T}(\rho)] = O(e^{-T}) + O(\epsilon_{stat}) + O(\epsilon_{bias})$ and an $\tilde{O}(1/\varepsilon^2)$ sample complexity.*

The optimality gap is bounded in this theorem via three terms, where the first term, related to the number of time steps $T$, is the optimization rate (linear convergence since $O(e^{-T})$), the second term $\epsilon_{stat} = \tilde{O}(\frac{1}{\sqrt{N}} + \frac{1}{\sqrt{K}})$ is a statistical error that depends on the number of samples $K, N$ in the robust critic and robust actor updates, and the last term $\epsilon_{bias}$ is the approximation error due to the limited representation power of value function approximation and the parameterized policy class. Omitting the approximation error $\epsilon_{bias}$, the sample complexities for achieving $\varepsilon$ robust optimal value are $\tilde{O}(1/\varepsilon^2)$, which achieves the optimal sample complexity in tabular setting [64]. However, RNAC with geometrically increasing step sizes induces a larger multiplicative constant factor (not shown in big-$O$ notation) and does not generalize well to general function approximation. We then analyze RNAC with a constant step size.

**Theorem 2** (Informal sublinear convergence of RNAC). *RNAC under the same specification as in Theorem 1 but with constant step size $\eta^t = \eta$ has $\mathbb{E}[V^{\pi^*}(\rho) - \frac{1}{T}\sum_{t=0}^{T-1} V^{\pi^t}(\rho)] = O(\frac{1}{T}) + O(\epsilon_{stat}) + O(\epsilon_{bias})$, implying an $\tilde{O}(1/\varepsilon^3)$ sample complexity.*

Although the theorem shows a slower optimization rate of RNAC with constant step size, this non-increasing step size is preferred in practice. Moreover, the analysis can be generalized to a general policy class with optimization rate $O(1/\sqrt{T})$, and an $\tilde{O}(1/\varepsilon^4)$ sample complexity.

**Algorithm 2: Robust Linear Temporal Difference (RLTD)**

**Input:** $\pi, K$
**Initialize:** $w_0, s_0$
**for** $k = 0, 1, \ldots, K-1$ **do**
    Sample $a_k \sim \pi(\cdot|s_k)$, $y_{k+1}$ according to $p^\circ_{s_k,a_k}$, and $s_{k+1}$ from $y_{k+1}$
    Update $w_{k+1} = w_k + \alpha_k \psi(s_k)\left[(\hat{\mathcal{T}}^\pi_{\mathcal{P}} V_{w_k})(s_k, a_k, y_{k+1}) - \psi(s_k)^\top w_k\right]$
    `// For DS:` $y_{k+1} = s'_{1:m} \overset{i.i.d.}{\sim} p^\circ_{s_k,a_k}$, $s_{k+1} \sim$ `Unif`$(y_{k+1})$ `and` $\hat{\mathcal{T}}^\pi_{\mathcal{P}}$ `in (3)`
    `// For IPM:` $y_{k+1} = s_{k+1} \sim p^\circ_{s_k,a_k}$ `and` $\hat{\mathcal{T}}^\pi_{\mathcal{P}}$ `in (6)`
**Return:** $w_K$

## 5 Robust Critic

The robust critic estimates the robust value function with access to samples from the nominal model. One may note that in many previous actor-critic analyses for canonical RL [61, 29, 9], the critic learns the $Q$ function, while realistic implementations in on-policy algorithms (e.g., proximal policy optimization (PPO)) treat the $V$ function as the target of the critic for training efficiency. We consider a robust critic that learns the robust $V$ function to align with such realistic implementations.

We present the **robust linear temporal difference (RLTD)** Algorithm 2, which is similar to the canonical linear-TD algorithm, but with an empirical robust Bellman operator. It iteratively performs the sampling and updating procedures. The sampling procedure differs for the uncertainty set by double sampling and IPM as shown in the comments in Algorithm 2. Using the samples, the parameter for the linear value function approximation $V_{w_k} = \Psi w_k$ is updated with step size $\alpha_k$. The following assumption is common [10, 9, 29]:

**Assumption 1** (Geometric mixing). *For any policy $\pi$, the Markov chain $\{s_k\}$ induced by applying $\pi$ in the nominal model $p^\circ$ is geometrically ergodic with a unique stationary distribution $\nu^\pi$.*

The update procedure essentially minimizes the Mean Square Projected Robust Bellman Error $\text{MSPRBE}^\pi(w) = \|\Pi^\pi(\mathcal{T}^\pi_{\mathcal{P}} V_w) - V_w\|^2_{\nu^\pi}$, where $\|V\|_{\nu^\pi} = \sqrt{\sum_s \nu^\pi(s)V(s)^2}$ is the $\nu^\pi$ weighted norm, and $\Pi^\pi = \Psi(\Psi^\top D^\pi \Psi)^{-1}\Psi^\top D^\pi$ is the weighted projection matrix with $D^\pi = \text{diag}(\nu^\pi) \in \mathbb{R}^{\mathcal{S}\times\mathcal{S}}$. The minimizer of $\text{MSPRBE}^\pi(w)$, denoted by $w^\pi$, is the unique solution of the projected robust Bellman equation $V_w = \Pi^\pi \mathcal{T}^\pi_{\mathcal{P}} V_w$, which is equivalent to $0 = \Psi^\top D^\pi(\mathcal{T}^\pi_{\mathcal{P}} \Psi w - \Psi w)$. RLTD is thus a stochastic approximation algorithm since the empirical operator $\hat{\mathcal{T}}^\pi_{\mathcal{P}}$ ((3) or (6)) is unbiased with $\mathbb{E}_{(s,a,y')\sim\nu^\pi\circ\pi\circ p^\circ}\left[\psi(s)\left[(\hat{\mathcal{T}}^\pi_{\mathcal{P}} V_w)(s, a, y') - \psi(s)^\top w\right]\right] = \Psi^\top D^\pi(\mathcal{T}^\pi_{\mathcal{P}} V_w - V_w)$.

To ensure that RLTD converges to the optimal linear approximation $V_{w^\pi}$, it is crucial that the projected robust Bellman operator $\Pi^\pi \mathcal{T}^\pi_{\mathcal{P}}$ be a contraction map with some $\beta < 1$.

**Definition 2.** $\Pi^\pi \mathcal{T}^\pi_{\mathcal{P}}$ *is a $\beta$-contraction w.r.t.* $\|\cdot\|_{\nu^\pi}$ *if* $\|\Pi^\pi \mathcal{T}^\pi_{\mathcal{P}} V - \Pi^\pi \mathcal{T}^\pi_{\mathcal{P}} V'\|_{\nu^\pi} \leq \beta\|V - V'\|_{\nu^\pi}$.

Unlike in linear TD for MDP [57], Tamar et al. [53] make an additional assumption (Assumption 2) to establish the contraction property of $\Pi^\pi \mathcal{T}^\pi_{\mathcal{P}}$ (Proposition 3) for RMDP.

**Assumption 2.** *There exists $\beta \in (0, 1)$ with $\gamma p_{s,a}(s') \leq \beta p^\circ_{s,a}(s') \forall s, s' \in \mathcal{S}, \forall a \in \mathcal{A}$, and $\forall p \in \mathcal{P}$.*

**Proposition 2** (Prop.3 in [53]). *Under Assumption 2, $\Pi^\pi \mathcal{T}^\pi_{\mathcal{P}}$ is a $\beta$-contraction w.r.t.* $\|\cdot\|_{v^\pi}$.

The implicit uncertainty set $\mathcal{P}$ of double-sampling satisfies Assumption 2 for small $\delta$, and thus guarantees contraction of $\Pi^\pi \mathcal{T}^\pi_{\mathcal{P}}$. For example, for $m = 2$ as in (4), a $\delta < \frac{1-\gamma}{2\gamma}$ is sufficient. However, simply taking a small radius $\delta$ is not a panacea for all uncertainty sets:

**Proposition 3.** *For any $f$-divergence and radius $\delta > 0$, there exists a geometrically mixing nominal model such that the $f$-divergence defined uncertainty set violates Assumption 2.*

On the other hand, Assumption 2, though well-accepted [53, 45, 41], may not be necessary. The proposed IPM uncertainty set relates robustness and regularization with an explicit formula for robust Bellman operator as in (6). The contraction behavior of $\Pi^\pi \mathcal{T}^\pi_{\mathcal{P}}$ for IPM uncertainty set can be established without Assumption 2:

**Lemma 1.** *For IPM uncertainty set with radius $\delta < \lambda_{\min}(\Psi^\top D^\pi \Psi)\frac{1-\gamma}{\gamma}$, there exists $\beta < 1$ that $\Pi^\pi \mathcal{T}_\mathcal{P}^\pi$ is a $\beta$-contraction mapping w.r.t. norm $\|\cdot\|_{v^\pi}$.*

Since the contraction of $\Pi^\pi \mathcal{T}_\mathcal{P}^\pi$ is obtained by RMDP under DS or IPM uncertainty sets with small radius $\delta$, we have the first finite sample guarantee of RLTD by recent advances in Markovian stochastic approximation [10]:

**Theorem 3** (Informal convergence of robust critic: Details in Appendix C). *RLTD with step sizes $\alpha_k = \Theta(1/k)$ satisfies $\mathbb{E}[\|w_K - w^\pi\|^2] = \tilde{O}(\frac{1}{K})$.*

## 6 Robust Natural Actor

The robust natural actor updates the policy parameter $\theta$ along an ascent direction that improves the value via preconditioning through the KL-divergence $\mathrm{KL}(p, q) := \langle p, \log(p/q)\rangle$. It has been well explored for natural policy gradient (NPG)-like algorithms, such as TRPO and PPO in canonical RL. The ascent direction is obtained by the policy gradient theorem in canonical MDP. We therefore first discuss the policy gradient for RMDP, where policy $\pi_\theta$ is differentiably parameterized by $\theta$.

The robust value $V_\mathcal{P}^{\pi_\theta}(\rho) =: J(\theta)$ is typically Lipschitz under proper parameterization [61], and is therefore differentiable a.e. by Rademacher's theorem [14]. Where it is not differentiable, a Fréchet supergradient $\nabla J$ of $J$ exists if $\limsup_{\theta' \to 0} \frac{J(\theta') - J(\theta) - \langle \nabla J(\theta), \theta' - \theta\rangle}{\|\theta' - \theta\|} \leq 0$ [25]. The following contains the policy gradient theorem for canonical RL as a special case:

**Lemma 2** (Policy supergradient). *For a policy $\pi = \pi_\theta$ that is differentiable w.r.t. parameter $\theta$,*

$$\nabla_\theta V^\pi(\rho) = \frac{\mathbb{E}_{s \sim d_\rho^{\pi,\kappa_\pi}} \mathbb{E}_{a \sim \pi_s}[Q^\pi(s,a)\nabla_\theta \log \pi(a|s)]}{1-\gamma} = \frac{\mathbb{E}_{s \sim d_\rho^{\pi,\kappa_\pi}} \mathbb{E}_{a \sim \pi_s}[A^\pi(s,a)\nabla_\theta \log \pi(a|s)]}{1-\gamma}$$

*is a Fréchet supergradient of $V^\pi(\rho)$, where $\kappa_\pi$ is the worst-case transition kernel w.r.t. $\pi$.*

We consider log-linear policies $\pi_\theta(a|s) = \frac{\exp(\phi(s,a)^\top \theta)}{\sum_{a'} \exp(\phi(s,a')^\top \theta)}$, where $\phi(s,a) \in \mathbb{R}^d$ is the feature vector and $\theta \in \mathbb{R}^d$ is the policy parameter. (The general policy class is treated in Appendix D).

In canonical RL, the training and testing environments follow the same nominal transition $p^\circ$. NPG updates the policy by $\theta \leftarrow \theta + \eta \mathcal{F}_\rho(\theta)^\dagger \nabla_\theta V_{p^\circ}^{\pi_\theta}(\rho)$, where $\mathcal{F}_\rho(\theta)^\dagger$ is the Moore-Penrose inverse of the Fisher information matrix $\mathcal{F}_\rho(\theta) := \mathbb{E}_{(s,a) \sim d_\rho^{\pi_\theta, p^\circ} \circ \pi_\theta}[\nabla_\theta \log \pi_\theta(a|s)(\nabla_\theta \log \pi_\theta(a|s))^\top]$. An "equivalent" Q-NPG was proposed in [3] to update the policy by $\theta \leftarrow \theta + \eta u'$, where $u' = \arg\min_u \mathbb{E}_{(s,a) \sim d_\rho^{\pi, p^\circ} \circ \pi}[(Q_{p^\circ}^\pi(s,a) - u^\top \phi(s,a))^2]$. Note that $u$ determines a Q value function approximation $Q^u(s,a) := \phi(s,a)^\top u$, which is compatible with the log-linear policy class [52]. Since $Q^\pi$ contains the information on the ascent direction as suggested by Lemma 2, the Q-NPG update can be viewed as inserting the best compatible Q-approximation $Q^{u'}$ for the policy.

We adopt the Q-NPG to robust RL and propose the **Robust Q-Natural Policy Gradient (RQNPG)** (Algorithm 3). Note that unlike canonical RL, where $Q_{p^\circ}^\pi$ can be estimated directly from a sample trajectory of executing $\pi$ in model $p^\circ$, the robust $Q^\pi$ is hard to estimate with samples from $p^\circ$. A value function approximation $V_w$ from the critic comes to help, as we can approximate the robust Q function via $Q_w(s,a) = r(s,a) + \inf_{p \in \mathcal{P}_{s,a}} p^\top V_w$, which exactly matches $Q^\pi$ if $V_w = V^\pi$. The RQNPG obtains information on $Q^\pi$ by first approximating it via a critic value $V_w$-guided function $Q_w$, and then estimating $Q_w$ by a policy-compatible robust Q-approximation $Q^u$.

As shown in Algorithm 3, RQNPG estimates a compatible Q-approximation $Q^{u_N}$ by iteratively performing sampling and updating procedures, where the sampling procedure follows that of the RLTD (Algorithm 2). The update procedure essentially approaches the minimizer $u_w^\pi := \arg\min_u \mathbb{E}_{(s,a) \sim \nu^\pi \circ \pi}[(Q_w(s,a) - u^\top \phi(s,a))^2]$ by stochastic approximation with step size $\zeta_n$ (stochastic gradient descent with Markovian data), since conditioned on $(s,a)$, $\hat{\mathcal{T}}_\mathcal{P}^\pi V_w$ ((3) or (6)) is an unbiased estimator for the Q function approximation $Q_w$.

Now we look at a specific update $\theta^{t+1} = \mathtt{RQNPG}(\theta^t, \eta^t, w^t, N)$ with $\zeta_n = \Theta(1/n)$, where $w^t = \mathtt{RLTD}(\pi_{\theta^t}, K)$. The following theorem shows an approximate policy improvement property of the RQNPG update:

**Algorithm 3: Robust Q-Natural Policy Gradient (RQNPG)**

**Input:** $\theta, \eta, w, N$
**Initialize:** $u_0, s_0$
**for** $n = 0, 1, \ldots, N-1$ **do**

  Sample $a_n \sim \pi_\theta(\cdot|s_n)$, $y_{n+1}$ according to $p^\circ_{s_k,a_k}$ and determine $s_{n+1}$ from $y'_{n+1}$

  Update $u_{n+1} = u_n + \zeta_n \phi(s_n, a_n) \left[ (\hat{\mathcal{T}}^\pi_\mathcal{P} V_w)(s_n, a_n, y_{n+1}) - \phi(s_n, a_n)^\top u_n \right]$

  `// For DS:` $y_{n+1} = s'_{1:m} \overset{i.i.d.}{\sim} p^\circ_{s_n,a_n}$, $s_{n+1} \sim \texttt{Unif}(y_{n+1})$ `and` $\hat{\mathcal{T}}^\pi_\mathcal{P}$ `in (3)`

  `// For IPM:` $y_{n+1} = s_{n+1} \sim p^\circ_{s_n,a_n}$ `and` $\hat{\mathcal{T}}^\pi_\mathcal{P}$ `in (6)`

**Return:** $\theta + \eta u_N$

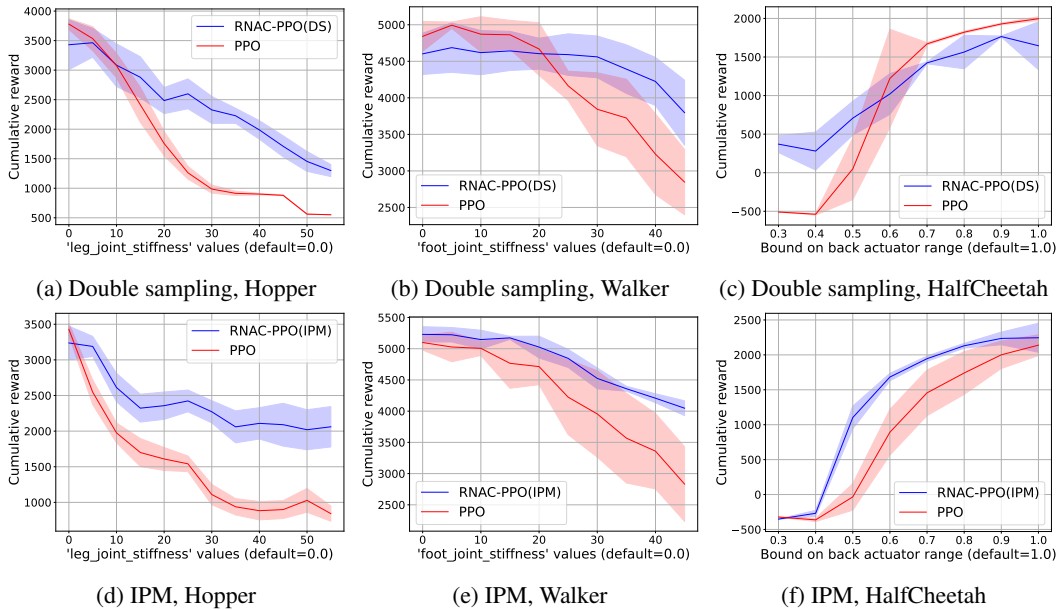

(a) Double sampling, Hopper     (b) Double sampling, Walker     (c) Double sampling, HalfCheetah

(d) IPM, Hopper     (e) IPM, Walker     (f) IPM, HalfCheetah

Figure 1: Cumulative rewards of RNAC-PPO(DS/IPM) and PPO on (a-c) stochastic MuJoCo Envs and (d-f) deterministic MuJoCo Envs under perturbation.

**Theorem 4** (Approximate policy improvement). *For any $t \geq 0$, we know*

$$V^{\pi^{t+1}}(\rho) \geq V^{\pi^t}(\rho) + \frac{\text{KL}_{d_\rho^{\pi^{t+1}, \kappa_{\pi^{t+1}}}}(\pi^t, \pi^{t+1}) + \text{KL}_{d_\rho^{\pi^{t+1}, \kappa_{\pi^{t+1}}}}(\pi^{t+1}, \pi^t)}{(1-\gamma)\eta^t} - \frac{\epsilon_t}{1-\gamma}, \quad (7)$$

*where* $\text{KL}_\nu(\pi, \pi') := \sum_s \nu(s) \text{KL}(\pi(\cdot|s), \pi'(\cdot|s)) \geq 0$ *and* $\mathbb{E}[\epsilon_t] = \tilde{O}(\frac{1}{\sqrt{N}} + \frac{1}{\sqrt{K}}) + O(\epsilon_{bias})$.

## 7 Experimental Results

We demonstrate the robustness of our RNAC approach (Algorithm 1) with Double-Sampling (DS) and IPM uncertainty sets on MuJoCo simulation environments [56]. We also perform real-world evaluations using TurtleBot [5], a mobile robot, on navigation tasks under action/policy perturbation. We implement a practical version RNAC using neural network function approximation, with the robust critic minimizing squared robust TD-error and the robust natural actor performing a robust proximal policy optimization (PPO) (see Algorithm 4 in Appendix A for details). We call this RNAC algorithm as RNAC-PPO and compare it with the canonical PPO algorithm [48]. Additional experimental results and details are deferred to Appendix A. We provide code with detailed instructions at https://github.com/tliu1997/RNAC.

### 7.1 MuJoCo Environments

We present the experimental results for perturbed MuJoCo Envs (Hopper-v3, Walker2d-v3 and HalfCheetah-v3) by changing their physical parameters (*leg_joint_stiffness*, *foot_joint_stiffness* and

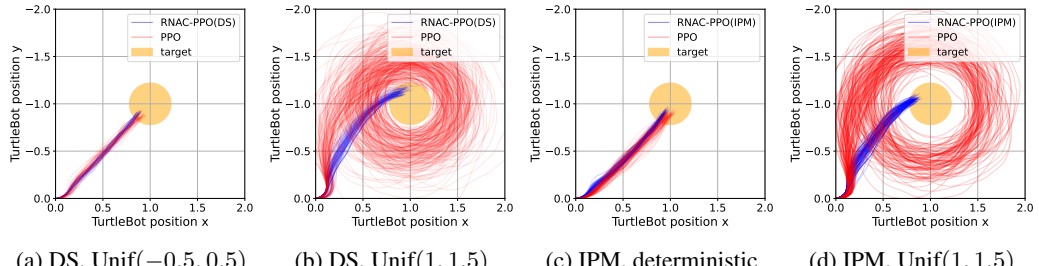

| (a) DS, Unif$(-0.5, 0.5)$ | (b) DS, Unif$(1, 1.5)$ | (c) IPM, deterministic | (d) IPM, Unif$(1, 1.5)$ |

Figure 2: (a-b) show trajectories under balanced (nominal) / unbalanced noise perturbed envs; (c-d) show trajectories under deterministic (nominal) / unbalanced noise perturbed envs.

*back_actuator_range*). We compare the performance of RNAC-PPO with that of the canonical PPO algorithm in Fig. 1, where the curves are averaged over 30 different seeded runs and the shaded region indicates the mean $\pm$ standard deviation. RNAC-PPO and PPO are trained with data sampled from the nominal models (e.g., *leg_joint_stiffness*$= 0.0$, *foot_joint_stiffness*$= 0.0$, and *back_actuator_range*$= 1.0$).

*DS Uncertainty Set:* MuJoCo environments have deterministic models. However, most uncertainty sets are only reasonable for stochastic models, e.g., $f$-divergence uncertainty sets. So, we add a uniform actuation noise $\sim$ Unif[-5e-3, 5e-3] in constructing stochastic MuJoCo environments as in [65]. We use $m = 2$ and radius $\delta = 1/6$ for RNAC-PPO in (4). The choice of $m = 2$ makes the training time of RNAC-PPO with DS uncertainty set and PPO comparable. Fig. 1a-1c demonstrate the robust performance of RNAC-PPO with Double-Sampling (DS) uncertainty sets in stochastic MuJoCo environments. Compared to PPO, the cumulative rewards of RNAC-PPO decay much slower as perturbations increase though they are slightly lower at the beginning (i.e., for the nominal model). The slight drop in initial performance may stem from optimizing the robust value function (1) under the worst-case transition models instead of the nominal models.

*IPM Uncertainty Set:* Since IPM with robust Bellman operator in (6) establishes robustness by negative regularization, it applies to environments with deterministic transition kernels. We select $\delta = 10^{-5}$ in (6) and evaluate RNAC-PPO with IPM uncertainty sets on deterministic MuJoCo environments in Fig. 1d-1f. RNAC-PPO has more robust behaviors with slow cumulative reward decay as perturbations increase. Notably, RNAC-PPO enjoys similar and sometimes even better initial performance on the nominal model compared to PPO, which we believe is due to the regularization of neural network parameters suggested by IPM (6) that can potentially improve neural network training.

### 7.2 TurtleBot Experiments

We demonstrate the robustness of the policy learned by RNAC-PPO on a real-world mobile robot (Fig. 3). We consider a navigation task as illustrated in Fig. 2, where the goal of the policy is to navigate the TurtleBot from the origin $(0, 0)$ to a target region centered at $(1, -1)$.

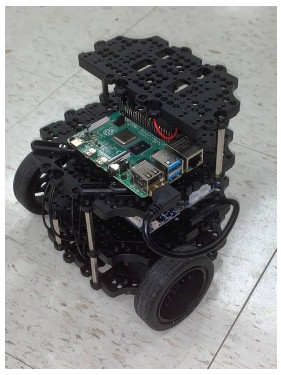

*DS Uncertainty Set:* We train RNAC-PPO and PPO on a stochastic nominal model with balanced actuation noise [54], and test the learned policies in the nominal model (Fig. 2a) and an unbalanced perturbed model (Fig. 2b). The policies learned by RNAC-PPO can reach the target region in both the nominal model and the perturbed model, while policies learned by PPO are fragile to perturbation and may not reach the target, as shown in Fig. 2b.

*IPM Uncertainty Set:* The robustness of the policies learned by RNAC-PPO trained on a deterministic nominal model is demonstrated in Fig. 2c and 2d, where the RNAC-PPO learned policies drive the robot to the target under perturbation, while the PPO-learned policies fail.

**A video of this real-world demonstration TurtleBot is available at [Video Link].**

Figure 3: TurtleBot Burger

## 8  Conclusion and Future Works

We have proposed two novel uncertainty sets based on double sampling and an integral probability metric, respectively, that are compatible with function approximation for large-scale robust RL. We propose a robust natural actor-critic algorithm, which to the best of our knowledge is the first policy-based approach for robust RL under function approximation with provable guarantees on learning the optimal robust policy. We demonstrate the robust performance of the proposed algorithm in multiple perturbed MuJoCo environments and a real-world TurtleBot navigation task.

Although several new theoretical and empirical results about large-scale robust RL are presented in this paper, there are still many open questions that need to be addressed. Current work focuses on the $(s, a)$-rectangular RMDP (Def. 1). We leave extensions to more general $s$-rectangular and non-rectangular RMDP for future works. Some theoretical analysis in this paper partly relies on Assumption 2. Though it is commonly made and accepted in theoretical works as discussed in paragraphs after Assumption 2, it is not a necessary condition. Further exploration of this can potentially lead to more theoretical advances. Another natural future direction is to extend current results to more complex settings such as robust constrained RL and robust multi-agent RL, following recent developments of policy gradient-based approaches in safe RL [68, 32] and multi-agent RL [67, 50].

## Acknowledgement

Dileep Kalathil's work is partially supported by the funding from the U.S. National Science Foundation (NSF) grant NSF-CAREER-EPCN-2045783.

P. R. Kumar's work is partially supported by the US Army Contracting Command under W911NF-22-1-0151 and W911NF2120064, US National Science Foundation under CMMI-2038625, and US Office of Naval Research under N00014-21-1-2385. The views expressed herein and conclusions contained in this document are those of the authors and should not be interpreted as representing the views or official policies, either expressed or implied, of the U.S. NSF, ONR, ARO, or the United States Government. The U.S. Government is authorized to reproduce and distribute reprints for Government purposes notwithstanding any copyright notation herein.

Portions of this research were conducted with the advanced computing resources provided by Texas A&M High Performance Research Computing.

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

# A Experimental Details and Additional Experimental Results

In this section, we provide details of the RNAC algorithm (Algorithm 1) implemented in the experiments – robust natural actor-critic proximal policy optimization (RNAC-PPO) algorithm (Algorithm 4). We also demonstrate further experimental results evaluated on different perturbations of physical hyperparameters in Hopper-v3, Walker2d-v3, and HalfCheetah-v3 from OpenAI Gym [7] compared with soft actor-critic (SAC) [18] and soft-robust [11] PPO (SRPPO). Finally, we introduce experimental details of the TurtleBot navigation task.

## A.1 RNAC-PPO Algorithm

We provide the RNAC-PPO algorithm in Algorithm 4, where the robust critic is minimizing squared robust TD error (MSRTDE) and the robust natural actor is performing the clipped version of robust PPO (RPPO) for computational efficiency, and we name it RNAC-PPO for simplicity. We implement RNAC-PPO employing neural network (NN) function approximation, where the robust natural actor is utilizing a neural Gaussian policy [48] with two hidden layers of width 64, and the value function in the robust critic is also parameterized by an NN with two hidden layers of width 64. Compared with the canonical empirical Bellman operator in the PPO algorithm, we adopt the robust empirical Bellman operator in the Robust PPO algorithm based on the double-sampling (DS) uncertainty set and the integral probability metric (IPM) uncertainty set, which can be efficiently computed. For a better comparison with SAC in the next subsection, we adopt several modifications (e.g., state normalization, reward scaling, gradient clip, etc) in the implementation of PPO-based algorithms to improve their performance under the nominal model. Note that for fairness we employ the same modification for both PPO and RNAC-PPO algorithms.

We use the same hyperparameters across different MuJoCo environments. Specifically, we select $\gamma = 0.99$ for the discount factor, $\eta^t = \alpha^t = 3 \times 10^{-4}$ for learning rates of both actor and critic updates implemented by ADAM [24], $T = 3 \times 10^6$ for the maximum training steps, and $B = 2048$ for the batch size.

---

**Algorithm 4: RNAC-PPO – MSRTDE + RPPO**

---

**Input:** $T, B, \eta^{0:T-1}, \alpha^{0:T-1}$
**Initialize:** $\theta^0$ for policy parameterization, $w^{-1}$ for value function approximation
**for** $t = 0, 1, \ldots, T-1$ **do**

    Collect set of trajectories $\mathcal{D}^t$ until $|\mathcal{D}^t| = B$ by running policy $\pi_{\theta^t}$ under $p^\circ$
    Update value function by minimizing mean-squared TD error with learning rate $\alpha^t$

$$w^t = \arg\min_w \frac{1}{|\mathcal{D}^t|} \sum_{(s,a,y') \in \mathcal{D}^t} \left[ V_w(s) - (\hat{\mathcal{T}}_{\mathcal{P}}^\pi V_{w^{t-1}})(s,a,y') \right]^2.$$

    // For DS: $y' = s'_{1:m} \overset{i.i.d.}{\sim} p^\circ_{s,a}$, $s' \sim \texttt{Unif}(y')$ and $\hat{\mathcal{T}}_{\mathcal{P}}^\pi$ in (3)
    // For IPM: $y' = s' \sim p^\circ_{s,a}$ and $\hat{\mathcal{T}}_{\mathcal{P}}^\pi$ in (6)
    Compute

$$\hat{A}^t(s,a) = (\hat{\mathcal{T}}_{\mathcal{P}}^\pi V_{w^t})(s,a,y') - V_{w^t}(s), \forall (s,a,y') \in \mathcal{D}_t$$

    Update policy via maximizing robust PPO objective with learning rate $\eta^t$

$$\theta^{t+1} = \arg\max_\theta \frac{1}{|D^t|} \sum_{(s,a,y') \in \mathcal{D}^t} \min \left( \frac{\pi_\theta(a|s)}{\pi_{\theta^t}(a|s)} \hat{A}^t(s,a), \text{clip}(\epsilon, \hat{A}^t(s,a)) \right),$$

    where $\text{clip}(\epsilon, A) = \begin{cases} (1+\epsilon)A, & \text{if } A \geq 0 \\ (1-\epsilon)A, & \text{if } A < 0 \end{cases}$

**Output:** $\theta^T$

---

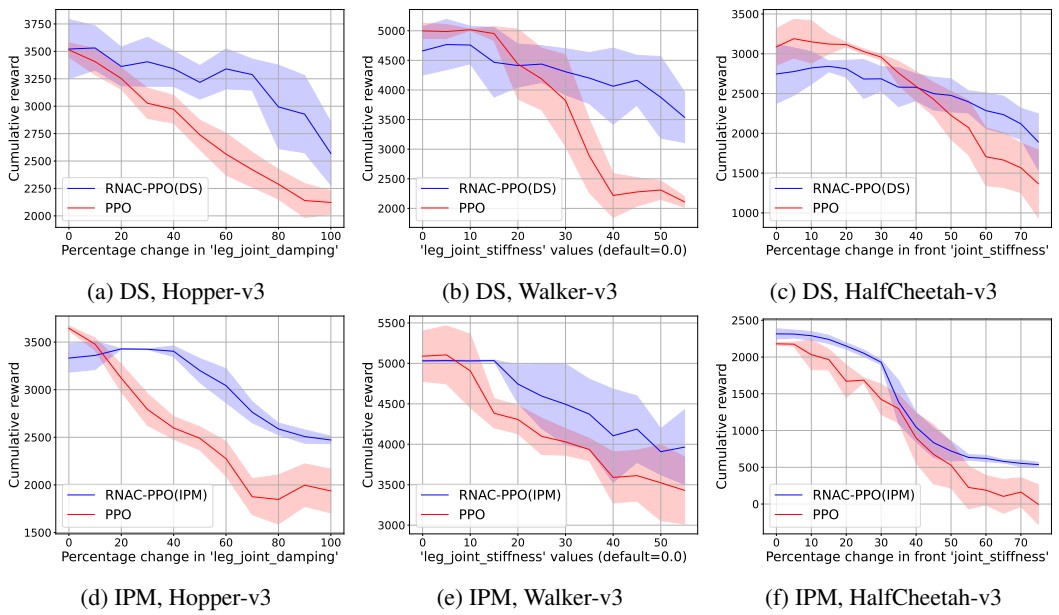

Figure 4: Cumulative rewards of RNAC-PPO(DS/IPM) and PPO on (a-c) stochastic MuJoCo Envs and (d-f) deterministic MuJoCo Envs under perturbation.

## A.2 Perturbed Mujoco Environments with Other Hyperparameters

In this subsection, we provide more experimental results on different perturbations of physical hyperparameters (e.g., *leg_joint_damping*, *leg_joint_stiffness*, and front *joint_stiffness*) in Hopper-v3, Walker2d-v3, and HalfCheetah-v3. Figure 4 shows that the RNAC-PPO algorithm is consistently robust compared to the PPO algorithm.

## A.3 Comparison with Soft Actor-Critic and Soft-Robust PPO

In Section 7, we demonstrate the robust behavior of the RNAC-PPO algorithm compared with the PPO algorithm in Figure 1. In this subsection, we add two more baselines for robust algorithms: soft actor-critic (SAC) [18] and soft-robust [11] PPO (SRPPO). SAC is regarded as one of the robust baselines since maximum entropy RL (e.g. SAC) was shown to solve some robust RL problems by maximizing the lower bound on a robust RL objective [13]. Soft-robust RL learns an optimal policy based on a distribution over an uncertainty set instead of considering the worst-case scenario [11]. For a fair comparison, we implement the idea of soft robustness into the framework of PPO. Specifically, we build an uncertainty set wrapping up four environments, including one nominal environment and three perturbed environments with *leg_joint_stiffness*=5.0 (default=0.0), *gravity*=-9.50 (default=-9.81), and *actuator_ctrlrange*=(-0.95, 0.95) (default=(-1.0, 1.0)), respectively. Additionally, we select [0.85, 0.05, 0.05, 0.05] as the distribution over the above uncertainty set for the SRPPO algorithm.

Figures 5c and 5d show that the cumulative rewards of RNAC-PPO decay much slower compared with those of PPO, but similar to those of SAC under the perturbation of *gravity*. This verifies the claim that SAC can solve some robust RL problems [13]. However, as shown in Figures 5a and 5b, SAC is not a panacea for all robust RL problems. Under the perturbation of *leg_joint_stiffness*, SAC suffers faster cumulative rewards decay compared with RNAC-PPO. Since SRPPO considers a distribution over an uncertainty set instead of only sampling from the nominal model, it doesn't perform well under the nominal model, but it shows a fairly robust behavior in Figures 5a and 5b when *leg_joint_stiffness* values are perturbed.

Since the cumulative rewards of SAC ($> 14000$) are much higher than those of PPO-based algorithms ($< 3000$) in HalfCheetah-v3, we only report the results of SAC in this subsection for Hopper-v3 to prevent the information of robustness from being blurred. Additionally, the training time of RNAC-PPO, PPO, and SRPPO is similar, which is at least 5 times less than that of SAC to

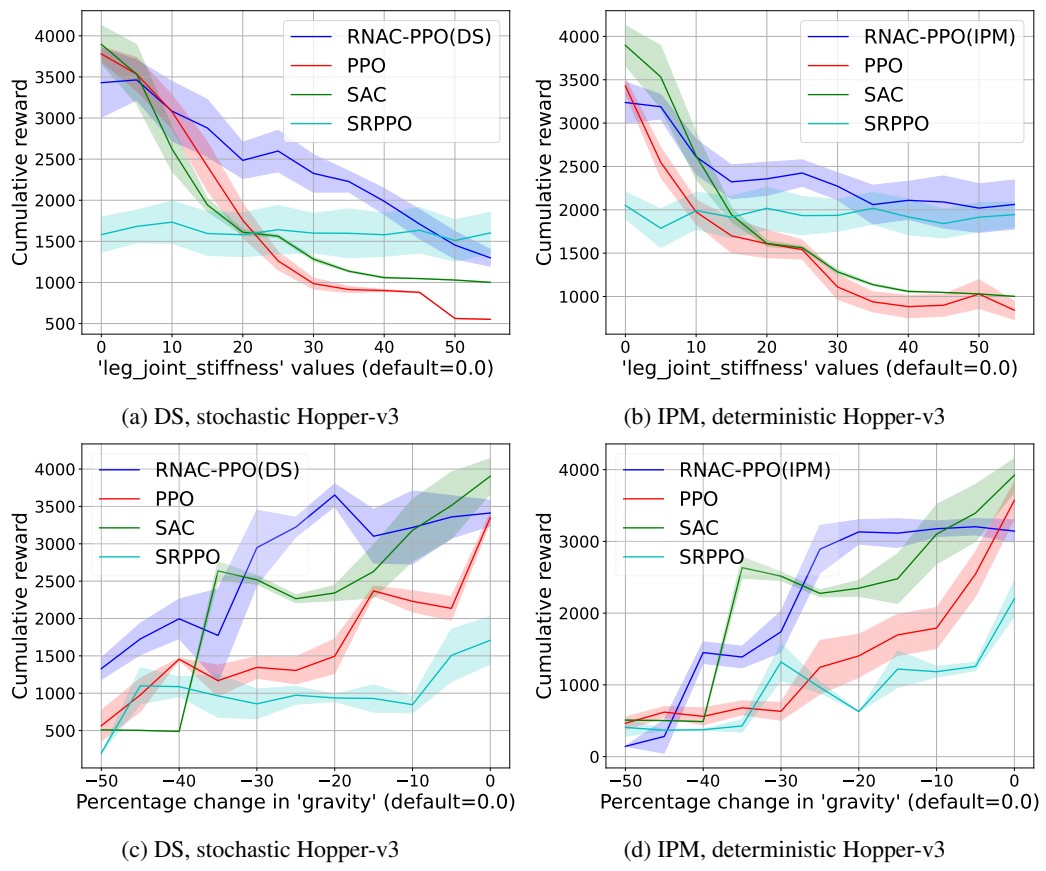

(a) DS, stochastic Hopper-v3          (b) IPM, deterministic Hopper-v3

(c) DS, stochastic Hopper-v3          (d) IPM, deterministic Hopper-v3

Figure 5: Cumulative rewards of RNAC-PPO(DS/IPM), PPO, SAC, and SRPPO on perturbed Hopper-v3 environments.

end the training of 3 million steps. This is due to the fact that PPO-based algorithms require fewer updates for critic $V$, while SAC requires more updates for critic Q.

### A.4 TurtleBot Experiment Details

The goal of this TurtleBot navigation task is to guide the robot to any desired target within a 2-meter range. The state space is a 2-dimensional continuous space, comprising the distance and relative orientation between the robot and the target. The robot is fixed to move towards its front with a linear velocity of 15 cm/s, while its angular velocity is controlled by the algorithm. Action space is 1-dimensional and continuous, ranging from $-2$ to $2$. The action signal is then linearly scaled into $[-1.5 \text{ rad/s}, 1.5 \text{ rad/s}]$. The reward function is designed to be proportional to the product of distance and action-scaled orientation between the robot and the target. Hitting the boundary or reaching the target would cause a reward of -200 or 200 respectively. One trajectory would end when the robot hits the boundary, reaches the goal, or the elapsed time is more than 150 seconds.

We train both PPO and RNAC-PPO(IPM) under the simulator Gazebo. To introduce stochasticity into the originally deterministic Gazebo environment, we apply uniform action noise perturbation. PPO and RNAC-PPO(DS) are trained under a balanced $\text{Unif}[-0.5, 0.5]$ noise-perturbed environment. All algorithms undergo 400 epochs of training, and the policy with the highest speed is saved. Subsequently, we evaluate all policies in an unbalanced $\text{Unif}[1.0, 1.5]$ noise-perturbed environment. We employ such a high noise since all learned algorithms are aggressive at fast turning, with action close to the limit of $-2$ or $2$. The trajectories of the robot under Gazebo simulator and a video for the real-world experiment are demonstrated in Figure 2 and **[Video Link]**.

Specifically, both PPO and RNAC-PPO are implemented by neural networks. The actor network is defined by the Gaussian policy, with one hidden layer and tanh activation function, projecting the

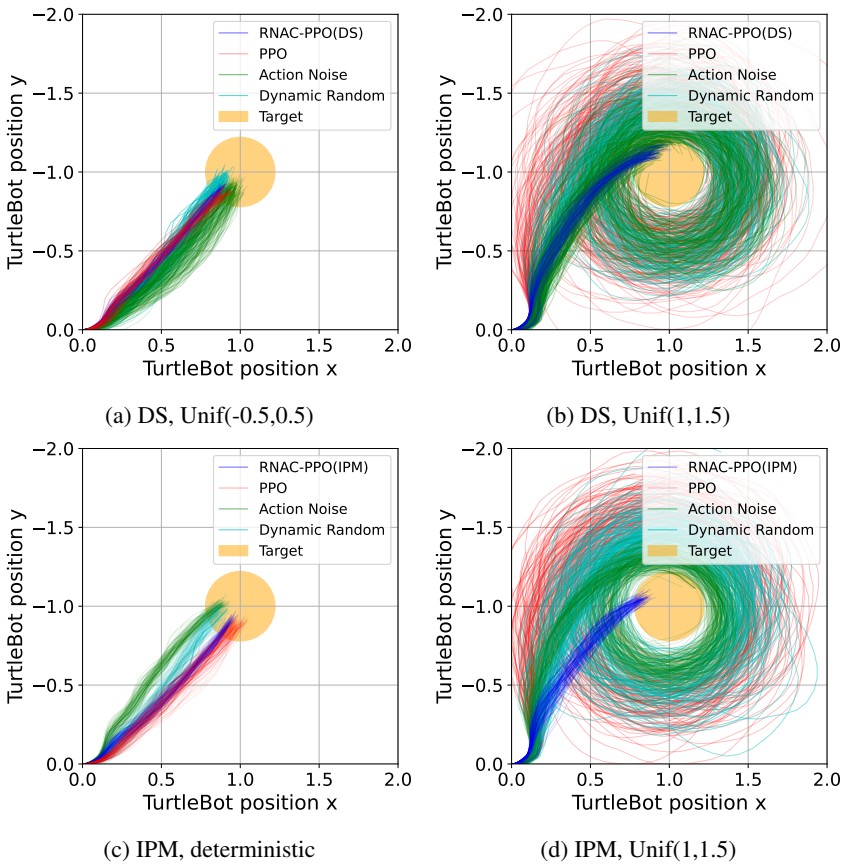

(a) DS, Unif(-0.5,0.5)  (b) DS, Unif(1,1.5)

(c) IPM, deterministic  (d) IPM, Unif(1,1.5)

Figure 6: Turtlebot's trajectories of RNAC-PPO(DS/IPM), PPO, dynamic randomization, and action noise envelope. (ac) are trajectories under a nominal environment, (bd) are trajectories under an unbalanced testing environment. In (b), target reaching rates are: 100% for RNAC-PPO(DS), 67.5% for action noise envelope, 9% for dynamic randomization, 0% for PPO. In (d), target reaching rates are: 100% for RNAC-PPO(IPM), 13% for action noise envelope, 2% for dynamic randomization, 0% for PPO

input state into a feature with dimension 100, then into the 1-dimensional action. Critic network consists of one hidden layer, with a width 100, and a ReLU activation function. We choose discount factor as 0.99, learning rate as $2.5 \times 10^{-4}$, batch size as 64, and optimizer ADAM.

Additionally, we also add more baselines (i.e., dynamics randomization [44] and action noise envelope [20]) to demonstrate the robustness of the proposed methods in the real-world TurtleBot environment. As shown in Figure 6, the proposed algorithm RNAC-PPO (DS / IPM) enjoys higher target reaching rates (100% / 100%), compared with action noise envelope (67.5% / 13%), dynamic randomization (9% / 2%), and PPO (0% / 0%), under perturbed testing environments, which illustrates the robustness of the RNAC-PPO algorithm.

We end this section by illustrating our hardware configurations. All experimental results are carried out on a Linux server with 48-core RTX 6000 GPUs, 48-core Intel Xeon 6248R CPUs, and 384 GB DDR4 RAM.

## B  Discussions on Uncertainty Sets

$R$-**contamination** For $R$-contamination [60], the uncertainty set is $\mathcal{P} = \otimes_{s,a}\mathcal{P}_{s,a}$, where $\mathcal{P}_{s,a} = \{Rq + (1-R)p^o_{s,a} : q \in \Delta_{\mathcal{S}}\}$. It can be shown that

$$\inf_{p \in \mathcal{P}_{s,a}} p^\top V = (1-R)(p^\circ_{s,a})^\top V + R \min_{s'} V(s'). \tag{8}$$

Its corresponding robust Bellman operator requires searching the entire state space $\mathcal{S}$ to calculate the minimum value in $V$, and thus intractable for large state space.

$\ell_{\mathsf{p}}$ **norm** [28] For any nominal model $p^\circ$, the uncertainty set is $\mathcal{P} = \otimes_{s,a}\mathcal{P}_{s,a}$ with $\mathcal{P}_{s,a} = \{p : \|q - p^\circ_{s,a}\|_{\mathsf{p}} \le \delta, \ \sum_{s'} q(s') = 1\}$, where $\|\cdot\|_{\mathsf{p}}$ is an $\ell_{\mathsf{p}}$ norm with its dual norm $\|\cdot\|_{\mathsf{q}}$ satisfying $\frac{1}{\mathsf{p}} + \frac{1}{\mathsf{q}} = 1$, and the domain of $q \in \Delta_{\mathcal{S}}$ is relaxed to hyperplane $\sum_{s'} q(s') = 1$ due to the difficulty in handling the boundary of $\Delta_{\mathcal{S}}$. It was shown [28] that

$$\inf_{p \in \mathcal{P}_{s,a}} p^\top V = (p^\circ_{s,a})^\top V - \min_{w} \|V - w\mathbf{1}\|_{\mathsf{q}}, \tag{9}$$

which requires solving a minimization problem via binary search. The minimum value may have a closed-form representation (c.f. Table 1 in [27]). For example, the minimum value is the average of $V$ for $\ell_2$ norm, the median of $V$ for $\ell_1$ norm, and the average peak of $V$ (i.e., the average of the maximum and minimum value of $V$) for $\ell_\infty$ norm. $\ell_{\mathsf{p}}$ norm-based uncertainty set is thus intractable in the large-scale scenario.

$f$**-divergence** Given a continuous strict convex function $f : \mathbb{R}_+ \to \mathbb{R}$ with $f(1) = 0$, $f$-divergence is defined by $\mathrm{d}_f(q, p) = \sum_s p(s) f\left(\frac{q(s)}{p(s)}\right)$. By distributionally robust optimization literature [12], we have

$$\inf_{p \in \mathcal{P}_{s,a}} p^\top V = \sup_{\lambda > 0, \eta \in \mathbb{R}} \mathbb{E}_{s'}\left[ -\lambda f^*\left(\frac{-V(s') - \eta}{\lambda}\right) - \lambda\delta - \eta \right], \tag{10}$$

where $f^*$ is the Fenchel conjugate of $f$, i.e., $f^*(y) = \sup_{x > 0}(yx - f(x))$. Given such a formula, the robust Bellman operator can be estimated given samples from the nominal model but requires the optimal dual variables $\lambda, \eta$ for each $(s, a)$ pair.

**Wasserstein distance** Given some metric $d(\cdot, \cdot)$ on state space $\mathcal{S}$, Wasserstein-$\sigma$ distance is defined for any $p, q \in \Delta_{\mathcal{S}}$ that

$$W_\sigma(p, q) := \big(\inf_{\mu \in U(p,q)} \mathbb{E}_{(s,s') \sim \mu}[d(s, s')^\sigma]\big)^{\frac{1}{\sigma}},$$

where $U(p, q)$ is the set of all couplings of $p$ and $q$. Wasserstein RMDP has uncertainty set $\mathcal{P} = \otimes_{s,a}\mathcal{P}_{s,a}$ with $\mathcal{P}_{s,a} = \{q : W_\sigma(q, p^\circ_{s,a}) \le \delta\}$ and we have

$$\inf_{p \in \mathcal{P}_{s,a}} p^\top V = \sup_{\lambda > 0} \sum_{s'} p^\circ_{s,a}(s') \inf_{\tilde{s}}[V(\tilde{s}) + \lambda(d(\tilde{s}, s')^\sigma - \delta)]. \tag{11}$$

The Wasserstein RMDP has been previously studied [1, 26]. Kuang et al. [26] gives a state disturbance view of Wasserstein RMDP, where the robust Bellman operator requires searching the worst-case state in the vicinity of the next state sample, i.e., $\inf_{\tilde{s}: \|\tilde{s} - s'\| \le \delta} V(\tilde{s})$. Although this approach can be applied to RMDP with large state space, the theoretical guarantee provided in [26] is only for policy iteration (planning problem) in the tabular setting (i.e., the robust Bellman operator is a contraction mapping under $\|\cdot\|_\infty$).

## B.1 Double-Sampling Uncertainty Sets

When action $a$ is taken at state $s$, the nominal model transits to the next state $s' \sim p^0_{s,a}$. It can be viewed as a double-sampling process that the next state $s'$ is generated by uniformly sampling one from $m$ states $s'_1, s'_2, \ldots, s'_m$ sampled i.i.d. according to $p^0_{s,a}$. The transition kernel in the DS uncertainty set at $(s, a)$-pair can be viewed as selecting the next state $s'$ in $\{s'_1, \ldots, s'_m\}$ according to a distribution $\alpha \in \Delta(m)$ that is perturbed from a uniform distribution $\mathrm{Unif}(m)$ and potentially depends on the samples $\{s'_1, \ldots, s'_m\}$. The DS uncertainty set is implicitly defined by a choice of $m$, divergence measure $\mathrm{d}(\cdot, \cdot)$ and radius $\delta > 0$.

**Bellman completeness for robust linear MDP under DS uncertainty set:** A Q function class $\mathcal{G} \subset \mathbb{R}^{\mathcal{S} \times \mathcal{A}}$ is said to be Bellman complete if it is closed, and for all $g \in \mathcal{G}$, $\mathcal{T}^*g$ also lies in $\mathcal{G}$, where

$$\mathcal{T}^*g(s, a) := r(s, a) + \gamma \sum_{s'} p_{s,a}(s') \max_a g(s', a).$$

Bellman completeness is critical for reinforcement learning. For RMDP with the Double-Sampling (DS) uncertainty set, the linear function approximation can satisfy Bellman completeness if the nominal model is a linear MDP. In linear MDP [22], we have

$$p_{s,a}^\circ(s') = \phi(s,a)^\top \boldsymbol{\mu}(s'), \quad r(s,a) = \phi(s,a)^\top \theta,$$

where $\boldsymbol{\mu}$ is a vector of (signed) measures over $\mathcal{S}$. Then for any $V \in \mathbb{R}^{\mathcal{S}}$

$$r(s,a) + \gamma \mathbb{E}_{s' \sim p_{s,a}^\circ}[V(s')] = r(s,a) + \gamma \int_{s'} \phi(s,a)^\top \boldsymbol{\mu}(s')V(s')ds'$$

$$= \phi(s,a)^\top \left( \theta + \gamma \int_{s'} \boldsymbol{\mu}(s')V(s')ds' \right) \in span(\Phi).$$

For the DS uncertainty set $\mathcal{P} = \otimes_{s,a} \mathcal{P}_{s,a}$ determined by $m, \mathrm{d}(\cdot,\cdot), \delta$, let $f_V(s'_{1:m}) := \inf_{\alpha \in \Delta_{[m]}:d(\alpha,\mathrm{Unif}([m])) \leq \delta} \sum_{i=1}^m \alpha_i V(s'_i)$, and $\bar{f}_V(s'_1) := \mathbb{E}_{s'_{2:m} \overset{i.i.d.}{\sim} p_{s,a}^\circ}[f_V(s'_{1:m})]$. It then follows that for any $V$

$$r(s,a) + \gamma \inf_{p \in \mathcal{P}_{s,a}} p^\top V = r(s,a) + \gamma \mathbb{E}_{s'_{1:m} \overset{i.i.d.}{\sim} p_{s,a}^\circ}[f_V(s'_{1:m})]$$

$$= \phi(s,a)^\top \theta + \gamma \mathbb{E}_{s'_1 \sim p_{s,a}^\circ}[\bar{f}_V(s'_1)] = \phi(s,a)^\top \left( \theta + \gamma \int_{s'} \boldsymbol{\mu}(s')\bar{f}_V(s')ds' \right) \in span(\Phi).$$

The Bellman completeness for DS-based RMDP with linear MDP nominal model is thus proved.

## B.2  Integral Probability Metric Uncertainty Sets

Under the linear value function approximation, $V_w(s) = \psi(s)^\top w$. Denote $\Psi \in \mathbb{R}^{\mathcal{S} \times d}$ as the feature matrix by stacking up $\psi(s)^\top$, and the value function approximation is a linear regressor $V_w = \Psi w \in \mathbb{R}^{\mathcal{S}}$. Let $\Psi = [\psi_1, \ldots \psi_d]$ with $\psi_i \in \mathbb{R}^{\mathcal{S}}$, without loss of generality, assume $\psi_1$ is an all-one vector, which corresponds to the bias term of the linear regressor, and $\Psi$ is full column rank.

We propose the IPM $\mathrm{d}_{\mathcal{F}}(p,q) = \sup_{f \in \mathcal{F}}\{p^\top f - q^\top f\}$ determined by function class

$$\mathcal{F} = \{\Psi\xi : \xi \in \mathbb{R}^d, \|\xi\| \leq 1\} \quad \text{restatement of Eq (5) in matrix form.}$$

The robust Bellman operator $\mathcal{T}_{\mathcal{P}}^\pi V$ (2) requires solving the optimization $\inf_{q \in \mathcal{P}_{s,a}} q^\top V$, which is equivalent to

$$\min_{q \in \Delta_{\mathcal{S}}} q^\top V \quad s.t. \sup_{f \in \mathcal{F}} q^\top f - (p_{s,a}^\circ)^\top f \leq \delta.$$

We relax the domain constraint $q \in \Delta_{\mathcal{S}}$ to $\sum_s q(s) = 1$ and define $\mathcal{P}_{s,a} = \{q : \mathrm{d}_{\mathcal{F}}(q, p_{s,a}^\circ) \leq \delta, \sum_s q(s) = 1\}$. This relaxation omits the boundary effect of $\Delta_{\mathcal{S}}$, which facilitates the following analysis. The relaxation is also made for $\ell_p$ norm-based uncertainty set [28], and one can argue that it does not introduce any relaxation error for $p_{s,a}^\circ(s') > 0, \forall s, a, s'$ and small $\delta$.

**Restatement of Proposition 1.** *For the IPM with $\mathcal{F}$ in (5), we have* $\inf_{q \in \mathcal{P}_{s,a}} q^\top V_w = (p_{s,a}^\circ)^\top V_w - \delta\|w_{2:d}\|$.

*Proof of Proposition 1.* Denote $u = q - p_{s,a}^\circ$. The $\inf_{q \in \mathcal{P}_{s,a}} q^\top V$ under the relaxation is

$$\min_u (p_{s,a}^\circ)^\top \Psi w + u^\top \Psi w, \quad s.t. \sup_{\|\xi\| \leq 1} u^\top \Psi \xi \leq \delta, \ u^\top \mathbf{1} = 0.$$

Since $\Psi$ is full column rank and $\psi_1$ is an all 1 vector, we know $\{\Psi^\top u : u^\top \mathbf{1} = 0, u \in \mathbb{R}^S\} = \{y \in \mathbb{R}^d : y_1 = 0\}$. The optimization problem can then be written as

$$\min_y (p_{s,a}^\circ)^\top \Psi w + y_{2:d}^\top w_{2:d}, \quad s.t. \sup_{\|\xi_{2:d}\| \leq 1} y_{2:d}^\top \xi_{2:d} \leq \delta.$$

The constraint is equivalent to $\|y_{2:d}\|_* \leq \delta$ and the optimal value can then be written as $(p_{s,a}^\circ)^\top V_w - \delta\|w_{2:d}\|$, which concludes the proof of Proposition 1. $\qquad\square$

The IPM has many merits due to its capability to take advantage of the geometry of the domain (state space) through the function class.

## C   Robust Critic Analysis

We analyze the robust critic component in the RNAC algorithm (Algorithm 1). This section would also be of independent interest for the robust policy evaluation problem, where one aims to estimate the robust value of some policy given Markovian data.

The first two subsections focus on the linear function approximation and prove the main theorems as stated in Section 5 of the main paper. The last subsection focuses on the general function approximation.

### C.1   Linear Robust Value Function Approximation

**Setting:** We aim to approximate the robust value function $V^\pi$ thorough linear function class

$$V_w(s) = \psi(s)^\top w, \quad \forall s \in \mathcal{S}. \tag{12}$$

The "optimal" linear value approximation $V_{w^\pi}$ is the solution of the projected robust Bellman equation i.e.,

$$\Pi^\pi \mathcal{T}_{\mathcal{P}}^\pi V_{w^\pi} = V_{w^\pi},$$

where the projection matrix is $\Pi^\pi = \Psi(\Psi^\top D^\pi \Psi)^{-1} \Psi^\top D^\pi$ with $D^\pi = \mathrm{diag}(\nu^\pi)$ and $\nu^\pi$ is the state stationary distribution of executing policy $\pi$ on the nominal model $p^\circ$.

**Definition 3** (Linear value approximation error). $\epsilon_{V,bias} := \sup_\pi \|\Pi^\pi V^\pi - V^\pi\|_{\nu^\pi}$.

$w^\pi$ is the solution of the projected robust Bellman equation $\Pi^\pi \mathcal{T}_{\mathcal{P}}^\pi V_w = V_w$. When $\Pi^\pi \mathcal{T}_{\mathcal{P}}^\pi$ is $\beta$-contraction w.r.t. $\|\cdot\|_{\nu^\pi}$, we have (according to Corollary 4 in [53])

$$\|V_{w^\pi} - V^\pi\|_{\nu^\pi} \le \frac{1}{1-\beta}\|\Pi^\pi V^\pi - V^\pi\|_{\nu^\pi} \le \frac{\epsilon_{V,bias}}{1-\beta}. \tag{13}$$

#### C.1.1   Contraction of Projected Robust Bellman Operator

The contraction property of the projected robust Bellman operator is the key to guaranteeing the convergence of linear TD algorithms. The contraction property can be guaranteed when Assumption 2 is satisfied, and we show that the DS uncertainty set with a small $\delta$ indeed satisfies this assumption.

We refer to a divergence $\mathrm{d}(\cdot,\cdot)$ as $C$-continuous at uniform distribution $\mathrm{Unif}([m])$, if $\{\alpha \in \Delta_{[m]} : \mathrm{d}(\alpha, \mathrm{Unif}([m])) \le \epsilon\} \subseteq \{\alpha \in \Delta_{[m]} : \|\alpha - \mathrm{Unif}([m])\|_\infty \le C\epsilon\}, \forall \epsilon > 0$.

**Proposition 4.** *Uncertainty set implicitly defined by double sampling with divergence* $\mathrm{d}(\cdot,\cdot)$ *that is $C$-continuous at uniform distribution* $\mathrm{Unif}([m])$ *and* $\delta < \frac{1-\gamma}{\gamma m C}$ *satisfies Assumption 2.*

*Proof.* For the uncertainty set $\mathcal{P}$ implicitly defined by double-sampling with robust Bellman operator 3

$$(\mathcal{T}_{\mathcal{P}}^\pi V)(s) := \mathbb{E}_{a \sim \pi(\cdot|s), s'_{1:m} \sim p^\circ_{s,a}} \left[ r(s,a) + \gamma \inf_{\alpha \in \Delta_{[m]} : \mathrm{d}(\alpha, \mathrm{Unif}([m])) \le \delta} \sum_{i=1}^m \alpha_i V(s'_i) \right].$$

Since $\mathrm{d}(\cdot,\cdot)$ is $C$-continuous at uniform distribution, take arbitrary $(s,a,s')$, we know

$$\sup_{q \in \mathcal{P}_{s,a}} q(s') = \mathbb{E}_{s'_{1:m} \sim p^\circ_{s,a}} \left[ \sup_{\alpha \in \Delta_{[m]} : \mathrm{d}(\alpha, \mathrm{Unif}(m)) \le \delta} \sum_{i=1}^m \alpha_i \mathbb{1}(s'_i = s') \right]$$

$$\le \mathbb{E}_{s'_{1:m} \sim p^\circ_{s,a}} \left[ \sup_{\alpha \in \Delta_{[m]} : \|\alpha - \mathrm{Unif}(m)\|_\infty \le C\delta} \sum_{i=1}^m \alpha_i \mathbb{1}(s'_i = s') \right]$$

$$\le \mathbb{E}_{s'_{1:m} \sim p^\circ_{s,a}} \left[ \sum_{i=1}^m (\frac{1}{m} + C\delta) \mathbb{1}(s'_i = s') \right] = p^\circ_{s,a}(s')(1 + mC\delta).$$

To guarantee $\gamma \sup_{q \in \mathcal{P}_{s,a}} q(s') \le \beta p^\circ_{s,a}$ with some $\beta < 1$, we know need $(1 + mC\delta) < \frac{1}{\gamma}$, which can be achieved by $\delta < \frac{1-\gamma}{\gamma m C}$. $\qquad\square$

The following theorem shows that choosing a small $\delta$ is not a panacea for the well-known $f$-divergence uncertainty set. Note that many well-known metrics, such as KL-divergence, total variation, and $\chi^2$-divergence are special cases of $f$-divergence.

**Proposition 5** (Restatement of Proposition 3). *For any $f$-divergence and radius $\delta > 0$, there exists a geometrically mixing nominal model such that the $f$-divergence defined uncertainty set violates Assumption 2.*

*Proof of Proposition 3.* Given a continuous strict convex function $f : \mathbb{R}_+ \to \mathbb{R}$ with $f(1) = 0$, the $f$-divergence is defined as $\mathrm{d}_f(q, p) = \sum_s p(s) f(\frac{q(s)}{p(s)})$.

Given a nominal model $p^\circ$ and a radius $\delta > 0$, the $f$-divergence-based uncertainty set is $\mathcal{P}$ with $\mathcal{P}_{s,a} = \{q : \mathrm{d}_f(q, p^\circ_{s,a}) \leq \delta\}$. For any $f$ and $\delta > 0$, consider a nominal model $p^\circ$, which has a uniform transition probability, i.e., transits to a uniformly and randomly selected next state at any state-action pair, except for $(s, a)$. The transition probability at $(s, a)$ is $p^\circ_{s,a} = (\alpha, \frac{1-\alpha}{|\mathcal{S}|-1}, \frac{1-\alpha}{|\mathcal{S}|-1}, \dots, \frac{1-\alpha}{|\mathcal{S}|-1})$, where $\alpha$ is some parameter to be determined later. It is clear that this nominal model $p^\circ$ is well mixed for any $\alpha \in [0, 1]$. Consider another model $q$ which coincides with $p^\circ$ except for states $(s, a)$ with $q_{s,a} = (\alpha/\gamma, \frac{1-\alpha/\gamma}{|\mathcal{S}|-1}, \frac{1-\alpha/\gamma}{|\mathcal{S}|-1}, \dots, \frac{1-\alpha/\gamma}{|\mathcal{S}|-1})$. Note that $\frac{q_{s,a}(1)}{p^\circ_{s,a}(1)} = \frac{1}{\gamma}$, thus if $q \in \mathcal{P}$, the Assumption 2 of $\beta < 1$ is violated. Since as $\alpha \to 0$, by the continuity of $f$ and $f(1) = 0$, we have

$$\mathrm{d}_f(q_{s,a}, p^\circ_{s,a}) = \sum_{s'} p^\circ_{s,a}(s') f(\frac{q_{s,a}(s')}{p^\circ_{s,a}(s')}) = \alpha f(1/\gamma) + (1-\alpha) f(\frac{1-\alpha/\gamma}{1-\alpha})$$

$$= \alpha f(1/\gamma) + (1-\alpha) f(1 - \frac{1/\gamma - 1}{1-\alpha}\alpha) \to 0 < \delta.$$

Therefore, with a sufficiently small $\alpha$, we have $q_{s,a} \in \mathcal{P}_{s,a}$ and clearly $q \in \mathcal{P}$. Thus there does not exist a universal choice of $\delta$ for $f$-divergence-based uncertainty set to guarantee Assumption 2. $\square$

We next prove Lemma 1, which shows the contraction of robust Bellman operator for IPM-based RMDP without Assumption 2.

*Proof of Lemma 1.* Let $P^\pi(s'|s) := \sum_a p^\circ_{s,\pi(a|s)}(s')$ be the state transition kernel of executing policy $\pi$ on the nominal model $p^\circ$. We can view $P^\pi \in \mathbb{R}^{\mathcal{S} \times \mathcal{S}}$ as a matrix. Recall $D^\pi = \mathrm{diag}(\nu^\pi)$, we have

$$\|\mathcal{T}^\pi_{\mathcal{P}} \Psi w - \mathcal{T}^\pi_{\mathcal{P}} \Psi w'\|_{\nu^\pi} = \left\| \gamma P^\pi \Psi w - \gamma \delta \|w_{2:d}\| \mathbf{1} - \gamma P^\pi \Psi w' + \gamma \delta \|w'_{2:d}\| \mathbf{1} \right\|_{\nu^\pi}$$

$$\leq \gamma \|\Psi w - \Psi w'\|_{\nu^\pi} + \gamma \delta \left| \|w\| - \|w'\| \right| \leq \gamma \|\Psi w - \Psi w'\|_{\nu^\pi} + \frac{\gamma \delta}{\lambda_{\min}(\Psi^\top D^\pi \Psi)} \|\Psi w - \Psi w'\|_{\nu^\pi}.$$

Since $\Pi^\pi$ is a non-expansion mapping w.r.t. $\|\cdot\|_{\nu^\pi}$, $\Pi^\pi \mathcal{T}^\pi_{\mathcal{P}}$ is a contraction mapping if $\gamma(1 + \frac{\delta}{\lambda_{\min}(\Psi^\top D^\pi \Psi)}) < 1$, which is equivalent to the condition $\delta < \lambda_{\min}(\Psi^\top D^\pi \Psi) \frac{1-\gamma}{\gamma}$.

$\square$

### C.1.2 Convergence of Robust Linear TD

As discussed in the paragraph under Assumption 1, robust linear TD (RLTD) is a stochastic approximation (c.f. Section F.3 for a brief overview of stochastic approximation) since empirical operator $\hat{\mathcal{T}}^\pi_{\mathcal{P}}$ ((3) or (6)) is unbiased with

$$\mathbb{E}_{(s,a,y') \sim \nu^\pi \circ \pi \circ p^\circ} \left[ \psi(s) \left[ (\hat{\mathcal{T}}^\pi_{\mathcal{P}} V_w)(s, a, y') - \psi(s)^\top w \right] \right] = \Psi^\top D^\pi (\mathcal{T}^\pi_{\mathcal{P}} V_w - V_w).$$

We can then let $F(w, x) = \psi(s) \left[ (\hat{\mathcal{T}}^\pi_{\mathcal{P}} V_w)(s, a, y') - \psi(s)^\top w \right]$ with $x = (s, a, y')$ and RLTD is solving the following equation

$$\bar{F}(w) := \Psi^\top D(\mathcal{T}^\pi_{\mathcal{P}} \Psi w - \Psi w) = 0,$$

through stochastic approximation:

$$w_{k+1} = w_k + \alpha_k F(w_k, x_k), \quad x_k = (s_k, a_k, y_{k+1}).$$

It is clear that $\{x_k\}$ is also a Markov chain with domain $\mathcal{X} = \mathcal{S} \times \mathcal{A} \times \mathcal{Y}$, $\mathcal{Y} = \mathcal{S}^m$ for the double-sampling uncertainty set with $m$ samples and $\mathcal{Y} = \mathcal{S}$ for the IPM uncertainty set.

**Theorem 5** (Formal statement of Theorem 3). *Take $\delta$ as in Lemma 1 and in Proposition 4, for the DS uncertain set or the IPM uncertainty set, respectively. Under Assumption 1, RLTD with step sizes $\alpha_k = \Theta(1/k)$ guarantees $\mathbb{E}[\|w_K - w^\pi\|^2] = \tilde{O}(\frac{1}{K})$ and $\mathbb{E}[\|V_{w_K} - V_{w^\pi}\|_{\nu^\pi}^2] = \tilde{O}(\frac{1}{K})$.*

*Proof of Theorem 5.* By Proposition 4 and Lemma 1, DS or IPM with small $\delta$ can guarantee $\Pi^\pi \mathcal{T}_{\mathcal{P}}^\pi$ is a $\beta$-contraction w.r.t. $\|\cdot\|_{\nu^\pi}$ for some $\beta < 1$. The theorem can be proved by applying Lemma 12. For this purpose, we only need to show that Assumption 8 is satisfied for RLTD. We check the three conditions in Assumption 8 as follows.

1. The geometric mixing property of $\{x_k = (s_k, a_k, y_{k+1})\}_{k \geq 0}$ is straightforward given geometrically mixed $\{s_k\}_{k \geq 0}$.

2. Since $r \in [0, 1]$ and $\|\psi(s)\|$ is bounded $\forall s \in \mathcal{S}$, we have $\|F(0, x)\| = \|\psi(s) r(s, a)\| \leq \|\psi(s)\|$ is also bounded for any $x = (s, a, y')$.

   For double-sampling (3) and any $x = (s, a, s'_{1:m})$,

   $$F(w, x) = \psi(s) \left( r(s, a) + \gamma \inf_{\alpha \in \Delta_{[m]} : d(\alpha, \mathrm{Unif}(m)) \leq \delta} \sum_{i=1}^m \alpha_i V_w(s'_i) - \psi(s)^\top w \right).$$

   Let $\Psi_{s'_{1:m}} \in \mathbb{R}^{m \times d}$ be the matrix by stacking up $\psi(s'_1)^\top, \ldots, \psi(s'_m)^\top$, we have

   $$\|F(w_1, x) - F(w_2, x)\|$$
   $$\leq \gamma \|\psi(s)\| \left| \inf_\alpha \alpha^\top \Psi_{s'_{1:m}} w_1 - \inf_\alpha \alpha^\top \Psi_{s'_{1:m}} w_2 \right| + \gamma \|\psi(s)\psi(s)^\top (w_1 - w_2)\|$$
   $$\leq \gamma \|\psi(s)\| |\sup_\alpha \alpha^\top \Psi_{s'_{1:m}} (w_1 - w_2)| + \gamma \|\psi(s)\|^2 \|w_1 - w_2\|$$
   $$\leq \max_{s'} 2\gamma \|\psi(s')\|^2 \|w_1 - w_2\|,$$

   where $\sup_\alpha, \inf_\alpha$ are taking in $\{\alpha \in \Delta_{[m]} : d(\alpha, \mathrm{Unif}([m])) \leq \delta\}$. Thus there exists $L_1 = \max(\max_s \|\psi(s)\|, 2\gamma \max_s \|\psi(s)\|^2)$ that guarantee $\|F(0, x)\| \leq L_1$ and $\|F(w, x) - F(w', x)\| \leq L_1 \|w - w'\|$ for the double-sampling RMDP.

   For IPM (6) and any $x = (s, a, s')$,

   $$F(w, x) = \psi(s) \left( r(s, a) + \gamma \psi(s')^\top w - \gamma \delta \|w_{2:d}\| - \psi(s)^\top w \right).$$

   We have

   $$\|F(w, x) - F(w', x)\| \leq \max_s \left( (\gamma + 1) \|\psi(s)\|^2 + \gamma \delta \|\psi(s)\| \right) \|w - w'\|,$$

   which also satisfies the second item of Assumption 8 by setting $L_1 = \max(\max_s \|\psi(s)\|, \max_s \left( (\gamma + 1) \|\psi(s)\|^2 + \gamma \delta \|\psi(s)\| \right))$.

3. Note that for small $\delta$, $\Pi^\pi \mathcal{T}_{\mathcal{P}}^\pi$ is a $\beta$-contraction w.r.t. $\|\cdot\|_{\nu^\pi}$. Since there exists $w^\pi$ with $\bar{F}(w^\pi) = 0$, let $D := D^\pi = \mathrm{diag}(\nu^\pi)$, we have

   $$\langle w - w^\pi, \bar{F}(w) \rangle = \langle w - w^\pi, \bar{F}(w) - \bar{F}(w^\pi) \rangle$$
   $$= \langle w - w^\pi, \Psi^\top D(\mathcal{T}^\pi \Psi w - \mathcal{T}^\pi \Psi w^\pi) \rangle - \|\Psi(w - w^\pi)\|_{\nu^\pi}^2$$
   $$= \langle \Psi^\top D\Psi(w - w^\pi), (\Psi^\top D\Psi)^{-1} \Psi^\top D(\mathcal{T}^\pi \Psi w - \mathcal{T}^\pi \Psi w^\pi) \rangle - \|\Psi(w - w^\pi)\|_{\nu^\pi}^2$$
   $$= \langle D^{1/2}\Psi(w - w^\pi), D^{1/2}\Psi(\Psi^\top D\Psi)^{-1} \Psi^\top D(\mathcal{T}^\pi \Psi w - \mathcal{T}^\pi \Psi w^\pi) \rangle - \|\Psi(w - w^\pi)\|_{\nu^\pi}^2$$
   $$\leq \|\Psi(w - w^\pi)\|_{\nu^\pi} \|\Pi^\pi \mathcal{T}_{\mathcal{P}}^\pi \Psi w - \Pi^\pi \mathcal{T}_{\mathcal{P}}^\pi \Psi w^\pi\|_{\nu^\pi} - \|\Psi(w - w^\pi)\|_{\nu^\pi}^2$$
   $$\leq -(1 - \beta)\|\Psi(w - w^\pi)\|_{\nu^\pi}^2 \leq -(1 - \beta)\lambda_{\min}(\Psi^\top D\Psi)\|w - w^\pi\|^2,$$

   where the first inequality is due to the Cauchy-Schwarz inequality and the second inequality is due to the $\beta$-contraction property of $\Pi^\pi \mathcal{T}_{\mathcal{P}}^\pi$.

In addition, we have

$$\mathbb{E}[\|V_{w_K} - V_{w^\pi}\|_{\nu^\pi}^2] = \mathbb{E}[(w_K - w^\pi)^\top (\Psi^\top D^{\pi^\pi} \Psi)(w_K - w^\pi)]$$
$$\leq \lambda_{max}(\Psi^\top D^\pi \Psi)\mathbb{E}[\|w_K - w^\pi\|^2] = \tilde{O}(1/K).$$

$\square$

## C.2 Extension of General Value Function Approximation

**Setting:** In the general function approximation setting, we consider a known finite and bounded function class $\mathcal{G} \subset \mathbb{R}^{\mathcal{S}}$ to fit the robust value function $V^\pi$, i.e.,

$$\mathcal{G} \subset \mathbb{R}^{\mathcal{S}}, \quad |\mathcal{G}| < \infty, \quad |g(s)| < \infty, \forall g \in \mathcal{G}, \forall s \in \mathcal{S}. \tag{14}$$

Note that IPM-based RMDP with empirical robust Bellman operator (6) only applies to linear function approximation. We implement it similarly by adding a negative regularization with neural network approximating the robust values, which also induces robust behavior of the learned policy as illustrated in the experiments (Section 7). However, it may not directly match any specific uncertainty set. We thus only consider the DS-based RMDP in this general value function approximation setting.

We define the robust Bellman error as follows.

**Definition 4** (Robust Bellman Error). $\epsilon_{g,bias} := \max_\pi \max_{g \in \mathcal{G}} \min_{g' \in \mathcal{G}} \|\mathcal{T}_{\mathcal{P}}^\pi g - g'\|_{\nu^\pi}$.

If $\epsilon_{g,bias} = 0$, then the robust value is realizable within the function class, i.e., $V^\pi \in \mathcal{G}$ for any $\pi \in \mathcal{G}$.

### C.2.1 Fitted Robust Value Evaluation Algorithm

We propose the fitted robust value evaluation (FRVE) in Algorithm 5, a robust version of the fitted value evaluation commonly used in offline RL. This algorithm first samples a batch of data from the nominal model, then select the last half as training data for analytical purpose without losing the order while the Markovian data are close to the stationary distribution due to geometric mixing Assumption 1, and iteratively solve for a better robust value approximation based on the current approximation $g_k$ and its robust value estimate $\hat{\mathcal{T}}_{\mathcal{P}}^\pi(g_k)$.

---
**Algorithm 5: Fitted Robust Value Evaluation (FRVE)**

**Input:** $\pi, K$
**Initialize:** $g_0, s_0$
**for** $k = 0, 1, \ldots, K - 1$ **do**
    Sample $a_k \sim \pi(\cdot|s_k)$, $y_{k+1}$ according to $p^\circ_{s_k,a_k}$, and $s_{k+1}$ from $y_{k+1}$
    // For DS: $y_{k+1} = s'_{1:m} \overset{i.i.d.}{\sim} p^\circ_{s_k,a_k}$, $s_{k+1} \sim$ Unif$(y_{k+1})$ and $\hat{\mathcal{T}}_{\mathcal{P}}^\pi$ in (3)
Let $\mathcal{D} = \{(s_k, a_k, y_{k+1})\}_{k=K/2}^{K-1}$ be the dataset
**for** $k = 0, 1, \ldots, K - 1$ **do**
    Update $g_{k+1} = \arg\min_{g \in \mathcal{G}} \frac{1}{|\mathcal{D}|} \sum_{(s,a,y') \in \mathcal{D}} \left( (\hat{\mathcal{T}}_{\mathcal{P}}^\pi g_k)(s, a, y') - g(s) \right)^2$.
**Return:** $g_K$

---

**Theorem 6** (Convergence of FRVE). *For DS RMDP with $\delta$ suggested by Proposition 4, $\mathcal{T}_{\mathcal{P}}^\pi$ is a $\beta$-contraction mapping for some $\beta < 1$. Under Assumption 7, the return of FRVE Algorithm 5 has*

$$\mathbb{E}[\|V^\pi - g_K\|_{\nu^\pi}] \leq \beta^K(G_{\max} + \frac{1}{1-\gamma}) + \frac{\epsilon_{g,stat}}{1-\beta} + \frac{\epsilon_{g,bias}}{1-\beta},$$

*where $G_{\max} := \max_{g \in \mathcal{G}}((\hat{\mathcal{T}}_{\mathcal{P}}^\pi g)(s, a, y'), \|\mathcal{T}_{\mathcal{P}}^\pi g\|_{\nu^\pi}, \|g\|_{\nu^\pi})$ and $\epsilon_{g,stat} = \tilde{O}(1/\sqrt{K})$. Thus $\mathbb{E}[\|V^\pi - g_K\|_{\nu^\pi}] = \tilde{O}(1/\sqrt{K}) + O(\epsilon_{g,bias})$.*

*Proof.* By the contraction mapping of $\mathcal{T}_{\mathcal{P}}^{\pi}$

$$
\begin{aligned}
\|V^{\pi} - g_K\|_{\nu^{\pi}} &\leq \|\mathcal{T}_{\mathcal{P}}^{\pi} V^{\pi} - \mathcal{T}_{\mathcal{P}}^{\pi} g_{K-1}\|_{\nu^{\pi}} + \|\mathcal{T}_{\mathcal{P}}^{\pi} g_{K-1} - g_K\|_{\nu^{\pi}} \\
&\leq \beta\|V^{\pi} - g_{K-1}\|_{\nu^{\pi}} + \|\mathcal{T}_{\mathcal{P}}^{\pi} g_{K-1} - g_K\|_{\nu^{\pi}} \\
&\leq \sum_{k=1}^{K} \beta^{K-k}\|\mathcal{T}_{\mathcal{P}}^{\pi} g_{k-1} - g_k\|_{\nu^{\pi}} + \beta^K\|V^{\pi} - g_0\|_{\nu^{\pi}} \\
&\leq \sum_{k=1}^{K} \beta^{K-k}\|\mathcal{T}_{\mathcal{P}}^{\pi} g_{k-1} - g_k\|_{\nu^{\pi}} + \beta^K(\|g_0\|_{\nu^{\pi}} + \frac{1}{1-\gamma}).
\end{aligned}
$$

Note that $\mathcal{T}_{\mathcal{P}}^{\pi} g_{k-1}$ is a target that $g_k$ is an approximation through MSE with Markovian data. If the data is stationary, i.e., $s_{K/2} \sim \nu^{\pi}$, applying Lemma 11 and the union bound over $\mathcal{G}$ (taking $\mathcal{T}_{\mathcal{P}}^{\pi} g$ as a target for each $g \in \mathcal{G}$), we know with probability at least $1 - \delta$,

$$
\|\mathcal{T}_{\mathcal{P}}^{\pi} g_{k-1} - g_k\|_{\nu^{\pi}} = O\left(G_{max}\sqrt{\frac{\log(|\mathcal{G}|) + \log(1/\delta)}{K}}\right) + O(\epsilon_{g,bias}), \quad \forall k = 1, 2, \ldots, K.
$$

Let $P_{\mu}$ be a distribution on $\mathcal{D}$ with $x_{K/2} \sim \mu$, and let $P_{\nu^{\pi}}$ be a distribution on $\mathcal{D}$ with $x_{K/2} \sim \nu^{\pi}$. Take $\mu$ as the true distribution of $x_{K/2}$ according to the data sampling process as in Algorithm 5. We know $\|\mu - \nu^{\pi}\|_{TV} = O(e^{-K})$ by Assumption 1, and thus

$$
\begin{aligned}
\mathbb{E}_{\mathcal{D} \sim P_{\mu}}[\|\mathcal{T}_{\mathcal{P}}^{\pi} g_{k-1} - g_k\|_{\nu^{\pi}}] &\leq 2G_{\max}\|P_{\mu} - P_{\nu^{\pi}}\|_{TV} + \mathbb{E}_{\mathcal{D} \sim P_{\nu^{\pi}}}[\|\mathcal{T}_{\mathcal{P}}^{\pi} g_{k-1} - g_k\|_{\nu^{\pi}}] \\
&= 2G_{\max}\|\mu - \nu^{\pi}\|_{TV} + \mathbb{E}_{\mathcal{D} \sim P_{\nu^{\pi}}}[\|\mathcal{T}_{\mathcal{P}}^{\pi} g_{k-1} - g_k\|_{\nu^{\pi}}] \\
&= O(G_{\max} e^{-K}) + O\left(G_{\max}\sqrt{\frac{\log(|\mathcal{G}|) + \log(K)}{K}}\right) + O(\epsilon_{g,bias}).
\end{aligned}
$$

The proof of the theorem is thus concluded. $\qquad\square$

## D   Robust Natural Actor Analysis

We analyze the robust natural component in the RNAC algorithm (Algorithm 1).

We first discuss the Fréchet supergradient of the robust value function in the first subsection. The second subsection focuses on the linear function approximation and proves the main theorems as stated in Section 5 of the main paper. The last subsection focuses on the general function approximation.

### D.1   Policy Gradient and Performance Difference

The robust value function $J(\theta) = V^{\pi_{\theta}}(\rho)$ is in general not differentiable. But since it is Lipschitz (w.r.t. to Lipschitz policy parameterization), it is differentiable almost everywhere according to Rademacher's theorem [14]. At the place not differentiable, Fréchet supergradient $\nabla J$ of $J$ is then defined as

$$
\limsup_{\theta' \to 0} \frac{J(\theta') - J(\theta) - \langle \nabla J(\theta), \theta' - \theta \rangle}{\|\theta' - \theta\|} \leq 0.
$$

When function $J$ is differentiable, $J$ at any point $\theta$ has a unique Fréchet supergradient, which is the gradient of $J$ at $\theta$.

*Proof of Lemma 2.* The Fréchet supergradient exists for tabular RMDP [29], which is

$$
[\nabla_{\pi} V^{\pi}(\rho)]_{s,a} = \frac{1}{1-\gamma} d_{\rho}^{\pi,\kappa_{\pi}}(s) Q^{\pi}(s,a), \quad \forall s, a,
$$

where $[\nabla_{\pi} V^{\pi}(\rho)]_{s,a}$ indicates the $(s, a)$ coordinate of vector $\nabla_{\pi} V^{\pi}(\rho)$. $\pi = \pi_{\theta}$ is a policy parameterized by $\theta$.

$$
\nabla_{\theta} V^{\pi_{\theta}}(\rho) = \sum_{s,a} [\nabla_{\pi} V^{\pi}(\rho)]_{s,a} \nabla_{\theta} \pi_{\theta}(a|s)
$$

$$= \frac{1}{1-\gamma} \mathbb{E}_{s \sim d_\rho^{\pi,\kappa_\pi}} \mathbb{E}_{a \sim \pi_s} [Q^\pi(s,a) \nabla_\theta \log \pi_\theta(a|s)]$$

$$= \frac{1}{1-\gamma} \mathbb{E}_{s \sim d_\rho^{\pi,\kappa_\pi}} \mathbb{E}_{a \sim \pi_s} [A^\pi(s,a) \nabla_\theta \log \pi_\theta(a|s)],$$

where the first relation is by the chain rule of supergradient. $\qquad\square$

## D.2 Linear Function Policy Approximation

**Setting:** This subsection considers the log-linear policy with

$$\pi_\theta(a|s) = \frac{\exp(\phi(s,a)^\top \theta)}{\sum_{a'} \exp(\phi(s,a')^\top \theta)}, \quad \forall (s,a) \in \mathcal{S} \times \mathcal{A}, \tag{15}$$

where $\phi(s,a) \in \mathbb{R}^d$ is some known feature vector and $\theta \in \mathbb{R}^d$ is the policy parameter. Let $\Phi \in \mathbb{R}^{|\mathcal{S}||\mathcal{A}| \times d}$ be the feature matrix by stacking up the feature vectors $\phi(s,a)^\top$.

Recall the discussion of the proposed RQNPG in Section 6. We approximate the robust Q function $Q^\pi$ via $Q_w(s,a) = r(s,a) + \inf_{p \in \mathcal{P}_{s,a}} p^\top V_w$ given a value function approximation $V_w$ from the robust critic, and then approximate $Q_w$ by a policy-compatible [52] robust Q-approximation $Q^u = \Phi u$. In other words, we project $Q_w$ onto $span(\Phi)$. Denote by $\Pi_\Phi^\pi := \Phi(\Phi^\top \text{diag}(\nu^\pi \circ \pi)\Phi)^{-1}\Phi^\top \text{diag}(\nu^\pi \circ \pi) \in \mathbb{R}^{|\mathcal{S}||\mathcal{A}| \times |\mathcal{S}||\mathcal{A}|}$ the projection matrix onto space $span(\Phi)$ w.r.t. norm $\|\cdot\|_{\nu^\pi \circ \pi}$. We then define the approximation error $\epsilon_{Q,bias}$ below. When realizable, i.e., $Q^\pi \in span(\Phi)$, the approximation error $\epsilon_{Q,bias} = 0$.

**Definition 5.** $\epsilon_{Q,bias} := \max_{\pi,\pi'} \max_{d=d_\rho^{\pi',\kappa_{\pi'}}, d_\rho^{\pi^*,\kappa_\pi} \text{ or } \nu^{\pi^*}} \|\Pi_\Phi^\pi Q^\pi - Q^\pi\|_{d \circ Unif}.$

We assume a finite relative condition number (Assumption 3) (similar to that in [3]). The relative condition number is not necessarily related to the size of the state space (details are shown in Remark 6.3 of [3]).

**Assumption 3.** $\max_{\pi,\pi'} \max_{d=d_\rho^{\pi',\kappa_{\pi'}}, d_\rho^{\pi^*,\kappa_\pi} \text{ or } \nu^{\pi^*}} \sup_u \frac{\|\Phi u\|_{d \circ Unif}}{\|\Phi u\|_{\nu^\pi \circ \pi}} \le \xi < \infty$ *for some* $\xi$.

Now we look at *a specific update* $\theta^{t+1} = \text{RQNPG}(\theta^t, \eta^t, w^t, N)$, where $w^t = \text{RLTD}(\pi_{\theta^t}, K)$.

### D.2.1 RQNPG One-Step Analysis – Robust Q Function Approximation

In this update $\theta^{t+1} = \text{RQNPG}(\theta^t, \eta^t, w^t, N)$, where $w^t = \text{RLTD}(\pi_{\theta^t}, K)$. RQNPG first approximates $Q_{w^t}(s,a) = r(s,a) + \gamma \inf_{p \in \mathcal{P}_{s,a}} p^\top V_{w^t} = \mathbb{E}[(\hat{\mathcal{T}}_\mathcal{P}^\pi V_{w^t})(s,a,y')|s,a]$ by $Q^u(s,a) = \phi(s,a)^\top u$, as the caculation of $u_N$ in Algorithm 3. Let

$$u_*^t := \arg\min_u \mathbb{E}_{(s,a) \sim \nu^t \circ \pi^t}[(Q_{w^t}(s,a) - Q^u(s,a))^2].$$

be the optimal approximation for the target $Q_{w^t}$. $u_*^t$ is approximated by stochastic approximation (c.f. Section F.3 for a brief overview of stochastic approximation) with a mean squared error loss

$$L(u; V_{w^t}, \pi) = \mathbb{E}_{(s,a,y') \sim \nu^\pi \circ \pi \circ p^\circ} \left[ \left( (\hat{\mathcal{T}}_\mathcal{P}^\pi V_{w^t})(s,a,y') - \phi(s,a)^\top u \right)^2 \right].$$

We know $u_*^t$ is the unique solution of

$$0 = -\nabla_u L(u; V_{w^t}, \pi) = \mathbb{E}_{(s,a,y') \sim \nu^\pi \circ \pi \circ p^\circ} \left[ \phi(s,a) \left( (\hat{\mathcal{T}}_\mathcal{P}^\pi V_{w^t})(s,a,y') - \phi(s,a)^\top u \right) \right].$$

Let $F(u,x)$ be the function inside the expectation with $x = (s,a,y')$, and $\bar{F}(u)$ be the negative gradient $-\nabla_u L(u; V, \pi)$. We then solve this stochastic zero point problem by stochastic approximation as in

$$u_{n+1} = u_n + \zeta_n \phi(s_n, a_n) \left[ (\hat{\mathcal{T}}_\mathcal{P}^\pi V_w)(s_n, a_n, y_{n+1}) - \phi(s_n, a_n)^\top u_n \right] = u_n + \zeta_n F(u_n, x_n),$$

where $x_n = (s_n, a_n, y_{n+1})$.

**Lemma 3** (Convergence of compatible Q function approximation (SGD with Markovian data)). *Under Assumption 1, RQNPG (Algorithm 3) with step sizes $\zeta_n = \Theta(1/n)$ guarantees $\mathbb{E}[\|u_N - u_*^t\|^2] = \tilde{O}(\frac{1}{N})$ and $\mathbb{E}[\|Q^{u_N} - Q^{u_*^t}\|_{\nu^t \circ \pi^t}^2] = \tilde{O}(\frac{1}{N})$.*

*Proof of Lemma 3.* The lemma is implied by Lemma 12. To see this, we only need to show that Assumption 8 is satisfied. We check the three conditions in Assumption 8 as follows.

1. The geometric mixing property of $\{x_k\}_{k\geq 0}$ is straightforward given geometrically mixed $\{s_k\}_{k\geq 0}$.

2. Since $\|\phi(s,a)\|$ is bounded $\forall (s,a) \in \mathcal{S} \in \mathcal{A}$, similar proof follows as in the proof of 5.

3. Since there exists $u_\pi$ with $\bar{F}(u_\pi) = 0$ and $L(u;V,\pi)$ is $\lambda_{min}(\mathbb{E}_{s,a}[\phi_{s,a}\phi_{s,a}^\top])$-strongly convex, we have

$$\langle u - u_\pi, \bar{F}(u)\rangle = \langle u - u_\pi, \bar{F}(u) - \bar{F}(u_\pi)\rangle$$
$$= -\langle u - u_\pi, \nabla L(u;V,\pi) - \nabla L(u^\pi;V,\pi)\rangle \leq -\lambda_{min}(\Sigma_{\nu^\pi\circ\pi})\|u - u_\pi\|^2,$$

where $\Sigma_{\nu^\pi\circ\pi} := \mathbb{E}_{s\sim\nu^\pi, a\sim\pi}[\phi_{s,a}\phi_{s,a}^\top]$.

Denote by $\Pi_\Phi^\pi := \Phi(\Phi^\top \text{diag}(\nu^\pi\circ\pi)\Phi)^{-1}\Phi^\top \text{diag}(\nu^\pi\circ\pi) \in \mathbb{R}^{|\mathcal{S}||\mathcal{A}|\times|\mathcal{S}||\mathcal{A}|}$ the projection matrix of function $Q \in \mathbb{R}^{|\mathcal{S}||\mathcal{A}|}$ onto matrix $\Phi \in \mathbb{R}^{|\mathcal{S}||\mathcal{A}|\times d}$ under norm $\|\cdot\|_{\nu^\pi\circ\pi}$. We know $Q^{u_*^t} = \Pi_\Phi^{\pi^t} Q_{w^t}$. We have

$$\mathbb{E}[\|Q^{u_N} - Q^{u_*^t}\|_{\nu^t\circ\pi^t}^2] = \mathbb{E}[(u_*^t - u_N)^\top(\Phi^\top \text{diag}(\nu^t\circ\pi^t)\Phi)(u_*^t - u_N)]$$
$$\leq \lambda_{max}(\Phi^\top \text{diag}(\nu^t\circ\pi^t)\Phi)\mathbb{E}[\|u_*^t - u_N\|^2] = \tilde{O}(1/N).$$

$\square$

For the update at step $t$, $\theta^{t+1} = \text{RQNPG}(\theta^t, \eta^t, w^t, N)$ and $w^t = \text{RLTD}(\pi_{\theta^t}, K)$, let $\pi^t = \pi_{\theta^t}$ be the policy at step $t$. Let $u^t = u_N$ ($u_N$ in Algorithm 3) and Lemma 3 above shows that $\mathbb{E}[\|Q^{u^t} - Q^{u_*^t}\|_{\nu^t\circ\pi^t}^2] = \tilde{O}(\frac{1}{N})$. This does not necessarily implies that $Q^{u^t}$ and $Q^{\pi^t}$ are close since the property of the critic returned $w^t$ is required. We measure the difference between $Q^{u^t}$ and $Q^{\pi^t}$ in the following lemma.

Let $w_*^t = w^{\pi^t}$ and $w^t = \text{RLTD}(\pi_{\theta^t}, K)$ is the output of RLTD at step $t$. Define $\epsilon_{V,stat} := \max_{t=0,1,...,T-1}\{\sqrt{\mathbb{E}[\|V_{w_*^t} - V_{w^t}\|_{\nu^t}^2]}\}$, and $\epsilon_{V,stat} = \tilde{O}(1/\sqrt{K})$ by Theorem 3.

**Lemma 4.** *Under the conditions in Theorem 5 and Assumption 3, there is some $\epsilon = \epsilon_{stat} + \epsilon_{bias}$ that for any $\pi, \pi'$ and any $d = d_\rho^{\pi',\kappa_{\pi'}}, d_\rho^{\pi^*,\kappa_{\pi^t}}$ or $\nu^{\pi^*}$,*

$$\left|\mathbb{E}\left[\mathbb{E}_{(s,a)\sim d\circ\pi}[Q^{u^t}(s,a) - Q^{\pi^t}(s,a)]\right]\right| \leq \epsilon, \tag{16}$$

*where the outside expectation is taken w.r.t. the randomness of the data collected in $w^t = \text{RLTD}(\pi_{\theta^t}, K)$ and $\theta^{t+1} = \text{RQNPG}(\theta^t, \eta^t, w^t, N)$. Moreover, $\epsilon_{stat} = \sqrt{|\mathcal{A}|}(\xi\beta\epsilon_{V,stat} + \xi\epsilon_{Q,stat})$ with $\epsilon_{V,stat} = \tilde{O}(\frac{1}{\sqrt{K}})$ and $\epsilon_{Q,stat} = \tilde{O}(\frac{1}{\sqrt{N}})$; $\epsilon_{bias} = \sqrt{|\mathcal{A}|}\left(\frac{\xi\beta}{1-\beta}\epsilon_{V,bias} + \epsilon_{Q,bias}\right)$ with $\epsilon_{V,bias}$ and $\epsilon_{Q,bias}$ in Definitions 3 and 5, respectively.*

*Proof of Lemma 4.* For any $d = d_s^{\pi^{t+1},\kappa^{t+1}}, d_\rho^{\pi^*,\kappa^t}$ or $\nu^{\pi^*}$ and for any $\pi$, we know for any $Q, Q'$

$$\left|\mathbb{E}\left[\mathbb{E}_{(s,a)\sim d\circ\pi}[Q(s,a) - Q'(s,a)]\right]\right| \leq \mathbb{E}\left[\sqrt{|\mathcal{A}|\mathbb{E}_{(s,a)\sim d\circ\text{Unif}}[(Q(s,a) - Q'(s,a))^2]}\right]$$
$$= \sqrt{|\mathcal{A}|}\mathbb{E}\left[\|Q - Q'\|_{d\circ\text{Unif}}\right].$$

To quantify the error between $Q^{u^t} - Q^{\pi^t}$, recall $Q^{u_*^t} = \Pi_\Phi^{\pi^t} Q_{w^t}$ and we decompose it into

$$Q^{u^t} - Q^{\pi^t} = (Q^{u^t} - Q^{u_*^t}) + (\Pi_\Phi^{\pi^t} Q_{w^t} - \Pi_\Phi^{\pi^t} Q_{w_*^t}) + (\Pi_\Phi^{\pi^t} Q_{w_*^t} - \Pi_\Phi^{\pi^t} Q^{\pi^t}) + (\Pi_\Phi^{\pi^t} Q^{\pi^t} - Q^{\pi^t}),$$

and bound each term respectively. We can transfer the norm within the space spanned by $\Phi$ via the assumption that $\|\Phi u\|_{d\circ\text{Unif}} \leq \xi\|\Phi u\|_{\nu^\pi\circ\pi}$. Note that the first three terms all lie in the $span(\Phi)$, we have the first term bounded by

$$\mathbb{E}\left[\|Q^{u^t} - Q^{u_*^t}\|_{d\circ\text{Unif}}\right] \leq \xi\mathbb{E}\left[\|Q^{u^t} - Q^{u_*^t}\|_{\nu^{\pi^t}\circ\pi^t}\right] \leq \xi\sqrt{\mathbb{E}\left[\|Q^{u^t} - Q^{u_*^t}\|_{\nu^{\pi^t}\circ\pi^t}^2\right]} \leq \xi\epsilon_{Q,stat},$$

for some $\epsilon_{Q,stat} = \tilde{O}(\frac{1}{\sqrt{N}})$ as in Lemma 3. The second term is bounded by

$$\mathbb{E}\left[\|\Pi_\Phi^{\pi^t} Q_{w^t} - \Pi_\Phi^{\pi^t} Q_{w_*^t}\|_{d\circ\mathrm{Unif}}\right] \leq \xi\mathbb{E}\left[\|\Pi_\Phi^{\pi^t} Q_{w^t} - \Pi_\Phi^{\pi^t} Q_{w_*^t}\|_{\nu^{\pi^t}\circ\pi^t}\right]$$

$$\leq \xi\mathbb{E}\left[\|Q_{w^t} - Q_{w_*^t}\|_{\nu^{\pi^t}\circ\pi^t}\right] \leq \xi\beta\mathbb{E}[\|V_{w^t} - V_{w_*^t}\|_{\nu^t}] \leq \xi\beta\sqrt{\mathbb{E}[\|V_{w^t} - V_{w_*^t}\|_{\nu^t}^2]} \leq \xi\beta\epsilon_{V,stat}$$

for some $\epsilon_{V,stat} = \tilde{O}(\frac{1}{\sqrt{K}})$ as in Theorem 5. The third term is bounded by

$$\mathbb{E}\left[\|\Pi_\Phi^{\pi^t} Q_{w_*^t} - \Pi_\Phi^{\pi^t} Q^{\pi^t}\|_{d\circ\mathrm{Unif}}\right] \leq \xi\mathbb{E}\left[\|\Pi_\Phi^{\pi^t} Q_{w_*^t} - \Pi_\Phi^{\pi^t} Q^{\pi^t}\|_{\nu^{\pi^t}\circ\pi^t}\right] \leq \xi\mathbb{E}\left[\|Q_{w_*^t} - Q^{\pi^t}\|_{\nu^{\pi^t}\circ\pi^t}\right]$$

$$\leq \xi\beta\mathbb{E}[\|V_{w_*^t} - V^t\|_{\nu^\pi}] \leq \xi\beta\frac{\mathbb{E}[\|\Pi^t V^t - V^t\|_{\nu^\pi}]}{1-\beta} \leq \frac{\xi\beta\epsilon_{V,bias}}{1-\beta},$$

where the last inequality is by Definition 3 and inequality (13). The last term is then bounded by Definition 5 $\mathbb{E}[\|\Pi_\Phi^{\pi^t} Q^{\pi^t} - Q^{\pi^t}\|_{d\circ\mathrm{Unif}}] \leq \epsilon_{Q,bias}$. We thus have

$$\left|\mathbb{E}\left[\mathbb{E}_{(s,a)\sim d\circ\pi}[Q^{u^t}(s,a) - Q^{\pi^t}(s,a)]\right]\right| \leq \sqrt{|\mathcal{A}|}\left(\xi\epsilon_{Q,stat} + \xi\beta\epsilon_{V,stat} + \frac{\xi\beta}{1-\beta}\epsilon_{V,bias} + \epsilon_{Q,bias}\right),$$

which concludes the proof. $\qquad\square$

### D.2.2 RQNPG One-step Analysis – Mirror Ascent Update

Now we look at the policy improvement of the update $\theta^{t+1} = \mathrm{RQNPG}(\theta^t, \eta^t, w^t, N)$ with $\zeta_n = \Theta(1/n)$ (Algorithm 3), where $w^t = \mathrm{RLTD}(\pi_{\theta^t}, K)$. Let $u^t = u_N$ ($u_N$ in Algorithm 3), and we know the RQNPG update is $\theta^{t+1} = \theta^t + \eta^t u^t$.

This RQNPG update is equivalent to a certain mirror ascent update. Specifically, recall $Q^{u^t}(s,a) = \phi(s,a)^\top u^t$ is the approximated robust Q function, the RQNPG update $\theta^{t+1} = \theta^t + \eta^t u^t$ is equivalent to [4]

$$\pi_s^{t+1} \leftarrow \arg\max_{\pi\in\Delta_\mathcal{A}}\{\eta^t\langle Q_s^{u^t}, \pi\rangle - \mathrm{KL}(\pi, \pi_s^t)\} \quad \forall s\in\mathcal{S}, \tag{17}$$

where we let $\pi_s := \pi(\cdot|s)$ and $Q_s := Q(s,\cdot)$ for simplicity. Note that this update can be viewed as a mirror descent step with KL-divergence as Bregman divergence. Given this mirror descent formulation of policy update in Eq (17), the pushback property indicates that for any policy $\pi$ (Eq (2) in [68]),

$$\eta^t\langle Q_s^{u^t}, \pi_s^{t+1}\rangle - \mathrm{KL}(\pi_s^{t+1}, \pi_s^t) \geq \eta^t\langle Q_s^{u^t}, \pi_s\rangle - \mathrm{KL}(\pi_s, \pi_s^t) + \mathrm{KL}(\pi_s, \pi_s^{t+1}),$$

which is equivalent to the following fundamental inequality

$$\eta^t\langle Q_s^{u^t}, \pi_s - \pi_s^t\rangle + \eta^t\langle Q_s^{u^t}, \pi_s^t - \pi_s^{t+1}\rangle \leq \mathrm{KL}(\pi, \pi_s^t) - \mathrm{KL}(\pi, \pi_s^{t+1}) - \mathrm{KL}(\pi_s^{t+1}, \pi_s^t). \tag{18}$$

**Restatement of Theorem 4** (Approximate policy improvement) *For any $t \geq 0$, we know*

$$V^{\pi^{t+1}}(\rho) \geq V^{\pi^t}(\rho) + \frac{\mathrm{KL}_{d_\rho^{\pi^{t+1},\kappa_{\pi^{t+1}}}}(\pi^t, \pi^{t+1}) + \mathrm{KL}_{d_\rho^{\pi^{t+1},\kappa_{\pi^{t+1}}}}(\pi^{t+1}, \pi^t)}{(1-\gamma)\eta^t} - \frac{\epsilon_t}{1-\gamma}, \tag{19}$$

*where* $\mathrm{KL}_\nu(\pi, \pi') := \sum_s \nu(s)\mathrm{KL}(\pi(\cdot|s), \pi'(\cdot|s)) \geq 0$ *and* $\mathbb{E}[\epsilon_t] = \tilde{O}(\frac{1}{\sqrt{N}} + \frac{1}{\sqrt{K}}) + O(\epsilon_{bias})$.

*Proof of Theorem 4.* Let $\kappa^{t+1} = \kappa_{\pi^{t+1}}$ be the worst-case transition kernel w.r.t. policy $\pi^{t+1}$, and let $\epsilon_t = \mathbb{E}_{s\sim d_\rho^{\pi^{t+1},\kappa^{t+1}}}[\langle Q_s^{\pi^t} - Q_s^{u^t}, \pi_s^{t+1} - \pi_s^t\rangle]$. We have

$$V^{\pi^{t+1}}(\rho) - V^{\pi^t}(\rho) \geq \frac{1}{1-\gamma}\mathbb{E}_{s\sim d_\rho^{\pi^{t+1},\kappa^{t+1}}}[\langle Q_s^{\pi^t}, \pi_s^{t+1} - \pi_s^t\rangle]$$

$$= \frac{1}{1-\gamma}\mathbb{E}_{s\sim d_\rho^{\pi^{t+1},\kappa^{t+1}}}[\langle Q_s^{u^t}, \pi_s^{t+1} - \pi_s^t\rangle] + \frac{1}{1-\gamma}\mathbb{E}_{s\sim d_\rho^{\pi^{t+1},\kappa^{t+1}}}[\langle Q_s^{\pi^t} - Q_s^{u^t}, \pi_s^{t+1} - \pi_s^t\rangle]$$

---

**Algorithm 6: Robust Natural Policy Gradient (RNPG)**

---

**Input:** $\theta, \eta, w, N$

**Initialize:** $u_0, s_0$, let $\pi = \pi_\theta$

**for** $n = 0, 1, \ldots, N - 1$ **do**

    Sample $a_n \sim \pi_\theta(\cdot|s_n)$, $y_{n+1}$ according to $p^\circ_{s_k, a_k}$ and determine $s_{n+1}$ from $y'_{n+1}$

    Update $u_{n+1} = (1 - \lambda)u_n +$

    $\zeta_n \nabla_\theta \log \pi_\theta(a_n|s_n) \left[ (\hat{\mathcal{T}}^\pi_\mathcal{P} V_w)(s_n, a_n, y_{n+1}) - V_w(s_n) - \nabla_\theta \log \pi_\theta(a_n|s_n)^\top u_n \right]$.

    `// For DS:` $y_{n+1} = s'_{1:m} \overset{i.i.d.}{\sim} p^\circ_{s_n, a_n}$, $s_{n+1} \sim$ `Unif`$(y_{n+1})$ `and` $\hat{\mathcal{T}}^\pi_\mathcal{P}$ `in (3)`

    `// For IPM:` $y_{n+1} = s_{n+1} \sim p^\circ_{s_n, a_n}$ `and` $\hat{\mathcal{T}}^\pi_\mathcal{P}$ `in (6)`

**Return:** $\theta + \eta u_N$

---

$$\geq \frac{\text{KL}_{d^{\pi^{t+1}, \kappa^{t+1}}_\rho}(\pi^t, \pi^{t+1}) + \text{KL}_{d^{\pi^{t+1}, \kappa^{t+1}}_\rho}(\pi^{t+1}, \pi^t)}{(1 - \gamma)\eta^t} - \frac{\epsilon_t}{1 - \gamma},$$

where the first inequality is by Lemma 8, and the last inequality is by taking $\pi = \pi^t$ in the fundamental inequality Eq (18), which implies

$$\eta^t \langle Q^{u^t}_s, \pi^{t+1}_s - \pi^t_s \rangle \geq \text{KL}(\pi^t_s, \pi^{t+1}_s) + \text{KL}(\pi^{t+1}_s, \pi^t_s).$$

$\epsilon_t$ can then be bounded by Lemma 4 with $\mathbb{E}[\epsilon_t] \leq 2\epsilon = 2\epsilon_{stat} + 2\epsilon_{bias}$, where $\epsilon_{stat}, \epsilon_{bias}$ are specified in Lemma 4 with $\epsilon_{stat} = \tilde{O}(\frac{1}{\sqrt{N}} + \frac{1}{\sqrt{K}})$. □

### D.3 Extension of General Function Approximation of Policy

**Setting:** In this subsection, we consider a general policy class of form

$$\left\{ \pi_\theta(a|s) = \frac{\exp(f_\theta(s, a))}{\sum_{a' \in \mathcal{A}} \exp(f_\theta(s, a'))} \mid \theta \in \mathbb{R}^d \right\}, \tag{20}$$

where $f_\theta$ is a differentiable function. This general policy class contains the log-linear policy class as a special case by $f_\theta(s, a) = \phi(s, a)^\top \theta$.

#### D.3.1 Robust Natural Policy Gradient with General Function Approximation

For the general policy class in Eq (20), we propose a Robust NPG (RNPG) algorithm.

This algorithm can be applied to RNAC (Algorithm 1) for the robust natural actor update. Now we look at a specific update $\theta^{t+1} = \text{RNPG}(\theta^t, \eta^t, w^t, N)$, where $w^t$ is the output of the robust critic at step $t$. Note that for the critic with general function approximation Eq. (14), we slightly abuse the notation by $V_{w^t} = g^t$, where $g^t$ is the output of the FRVE (Algorithm 5) at step $t$ of RNAC. We can view $g \in \mathcal{G}$ is parameterized by some $w$, as indicated in the RNAC algorithm Algorithm 1.

Denote by $\phi^\theta(s, a) := \nabla_\theta \log \pi_\theta(a_n|s_n)$ and $\Phi^\theta \in \mathbb{R}^{|\mathcal{S}||\mathcal{A}| \times d}$ as the feature matrix stacking up feature vector $\phi^\theta$. For each $t = 0, 1, \ldots, T - 1$, we let $\phi^t := \phi^{\theta^t}$ and $\Phi^t := \Phi^{\theta^t}$ for simplicity. The RNPG update is $\theta^{t+1} \leftarrow \theta^t + \eta^t u^t$, where $u^t = u_N$ as the output of the stochastic gradient descent in Algorithm 6,

$$u_{n+1} = (1 - \lambda)u_n + \zeta_n \nabla_\theta \log \pi_{\theta^t}(a_n|s_n) \left[ (\hat{\mathcal{T}}^{\pi^{\theta^t}}_\mathcal{P} V_{w^t})(s_n, a_n, y_{n+1}) - V_{w^t}(s_n) - \nabla_\theta \log \pi_\theta(a_n|s_n)^\top u_n \right].$$

RNPG approximates the value approximated advantage function $A_{w^t}(s, a) := r(s, a) + \gamma \inf_{p \in \mathcal{P}_{s,a}} p^\top V_{w^t} - V_{w^t}(s)$ by $A^u_t := \Phi^t u$. Note that $(\hat{\mathcal{T}}^{\pi^t}_\mathcal{P} V_{w^t})(s_n, a_n, y_{n+1}) - V_{w^t}(s_n)$ is an unbiased estimate of $A_{w^t}(s_n, a_n)$, i.e.,

$$\mathbb{E}[(\hat{\mathcal{T}}^{\pi^t}_\mathcal{P} V_{w^t})(s_n, a_n, y_{n+1}) - V_{w^t}(s_n)|s_n, a_n] = A_{w^t}(s_n, a_n).$$

RNPG thus is iteratively solving the following optimization

$$\frac{1}{2} \|A_{w^t} - \Phi^t u\|^2_{\nu^t \circ \pi^t} + \frac{\lambda}{2} \|u\|^2, \tag{21}$$

by stochastic approximation (stochastic gradient descent with Markovian data). Denote by $u_*^{\pi^t}$ the optimal value of the optimization, which is also a solution of the equation

$$
\begin{aligned}
0 &= (\Phi^t)^\top (A_{w^t} - \Phi^t u) - \lambda u \\
&= \mathbb{E}_{(s,a,y') \sim \nu^\pi \circ \pi \circ p^\circ} \left[ \phi^t(s,a) \left( (\hat{\mathcal{T}}_{\mathcal{P}}^{\pi^t} V_{w^t})(s,a,y') - V_{w^t}(s) - \phi^t(s,a)^\top u \right) \right] - \lambda u.
\end{aligned}
$$

**Theorem 7** (Convergence of compatible advantage function approximation (SGD with Markovian data)). *Under assumption 1, RNPG($\theta^t, \eta^t, w^t, N$) (Algorithm 6) with step sizes $\zeta_n = \Theta(1/n)$ guarantees $\mathbb{E}[\|u_N - u_*^t\|^2] = \tilde{O}(\frac{1}{N})$ and $\mathbb{E}[\|A_t^{u_N} - A_t^{u_*^t}\|_{\nu^t \circ \pi^t}^2] = \tilde{O}(\frac{1}{N})$.*

*Proof of Theorem 7.* The theorem can be proved in the same manner as that for Lemma 3, since the objective function in Eq (21) is strongly convex. $\qquad\square$

With slight abuse of notation, denote by $\Sigma_{\nu,\pi}^t := \mathbb{E}_{(s,a) \sim \nu \circ \pi} \left[ \phi^t(s,a) \phi^t(s,a)^\top \right]$. The optimal value $u_*^t = (\lambda I + (\Phi^t)^\top \text{diag}(\nu^t \circ \pi^t) \Phi^t)^{-1} ((\Phi^t)^\top \text{diag}(\nu^t \circ \pi^t) A_{w^t})$ satisfies $\|u_*^t\| \leq \frac{\|A_{w^t}\|_{\nu^t \circ \pi^t} \sqrt{\sum_{i=1}^d \|\phi_i^t\|_{\nu^t \circ \pi^t}^2}}{\lambda + \lambda_{\min}(\Sigma_{\nu^t \circ \pi^t}^t)}$. Let $u^t = u_N$, Theorem 7 gives $\mathbb{E}[\|u^t\|^2] \leq 2\mathbb{E}[\|u_*^t\|^2] + \tilde{O}(\frac{1}{N})$. We then have $\mathbb{E}[\|u^t\|^2] \leq U$ under the following assumption.

**Assumption 4** (Bounded feature). *Assume $\sup_\theta \sum_{i=1}^d \|\phi_i^\theta\|_{\nu^{\pi_\theta} \circ \pi_\theta}^2 < \infty$. Since $\mathcal{G}$ (14) is bounded, $\|A_{w^t}\|_{\nu^t \circ \pi^t} < \infty$ and there exists $0 < U < \infty$ that $\max_t \mathbb{E}[\|u^t\|^2] \leq U$.*

### D.3.2 Robust General Advantage Function Approximation

Denote by $\Pi_{\Phi^t}^t$ the projection mapping to the space $\{\Phi^t u : \|u\| \leq \|u_*^t\|\}$ under metric $\|\cdot\|_{\nu^t \circ \pi^t}$.

**Definition 6.** $\epsilon_{A,bias} := \max_{t=0,1,\dots,T-1} \mathbb{E}\left[ \|\Pi_{\Phi^t}^t A^t - A^t\|_{\nu^{\pi^*} \circ \pi^*} \right]$.

Note that $\epsilon_{A,bias}$ also implicitly depends on the choice of $\lambda$ since $u_*^t$ depends on $\lambda$. If the realizable case, i.e., $A^{\pi^t} \in \text{span}(\Phi^t)$, $\epsilon_{A,bias} = 0$ if $\lambda = 0$.

**Assumption 5.** *There is some $0 < \xi' < \infty$ that for any $\theta$, $\sup_u \frac{\|\Phi^\theta u\|_{\nu^{\pi^*} \circ \pi^*}}{\|\Phi^\theta u\|_{\nu^{\pi_\theta} \circ \pi_\theta}} \leq \xi'$.*

**Lemma 5.** *Under the conditions in Theorems 7 and 6 and Assumption 5, for any $t$,*

$$
\left| \mathbb{E}\left[ \mathbb{E}_{(s,a) \sim \nu^{\pi^*} \circ \pi^*} [A^{u^t}(s,a) - A^{\pi^t}(s,a)] \right] \right| \leq \epsilon' = \epsilon'_{stat} + \epsilon'_{bias}, \tag{22}
$$

*where the outside expectation is taken w.r.t. the randomness of the data collected the robust natural actor and robust critic update. Moreover, $\epsilon'_{stat} = \tilde{O}(\frac{1}{\sqrt{K}} + \frac{1}{\sqrt{N}})$; $\epsilon'_{bias} = O(\epsilon_{A,bias} + \epsilon_{g,bias})$ or $O(\epsilon_{A,bias} + \epsilon_{V,bias})$ for linear robust critic.*

*Proof of Lemma 5.* To quantify the error between $A^{u^t} - A^t$, we decompose it into

$$
A^{u^t} - A^t = (A^{u^t} - A^{u_*^t}) + (\Pi_{\Phi^t}^t A_{w^t} - \Pi_{\Phi^t}^t A_{w_*^t}) + (\Pi_{\Phi^t}^t A_{w_*^t} - \Pi_{\Phi^t}^t A^t) + (\Pi_{\Phi^t}^t A^t - A^t),
$$

and bound each term respectively. The proof follows similarly to that in Lemma 4 by applying Theorem 7 and Theorem 6. $\qquad\square$

## E  Robust Natural Actor-Critic Analysis

In this section, we first state and prove the formal versions of the main theorems Theorem 1 and Theorem 2 in Theorem 8 and Theorem 9, respectively. We then give the convergence of RNAC employing general function approximation in Theorem 10.

## E.1 Linear Function Approximation

We introduce the detailed setup of the RNAC algorithm with linear function approximation, based on which the theorems are stated and proved.

**RNAC-Linear Setting:** We study the RNAC (Algorithm 1) with robust critic performing RLTD (Algorithm 2 with $\xi_n = \Theta(1/k)$ as in Theorem 5) and robust natural actor performing RQNPG (Algorithm 3 with $\zeta_n = \Theta(1/n)$ as in Lemma 3) under linear value function approximation as in Eq (12) and log-linear policy as in Eq (15), for DS or IPM uncertainty sets taking $\delta$ suggested by Proposition 4 or Lemma 1, respectively. We assume Assumption 1 and Assumption 3 hold.

For each time $t = 0, 1, \ldots, T - 1$, the value approximation and policy in RNAC are updated as $w^t = \mathtt{RLTD}(\pi_{\theta^t}, K)$ and $\theta^{t+1} = \mathtt{RQNPG}(\theta^t, \eta^t, w^t, N)$, respectively. Denote by $u^t$ as $u_N$ in $\mathtt{RQNPG}(\theta^t, \eta^t, w^t, N)$ (Algorithm 3), which defines a robust Q-function approximation $Q^{u^t}$. Let $\pi^t = \pi_{\theta^t}, \kappa^t = \kappa_{\pi^t}, V^t = V^{\pi^t}$ for simplicity. We have the following lemma.

**Lemma 6.** *Under the **RNAC-Linear Setting**, for any $t = 0, 1, \ldots, T-1$, we have $\langle Q_s^{u^t}, \pi_s^{t+1} - \pi_s^t \rangle \geq 0, \forall s \in \mathcal{S}$ and*

$$\mathbb{E}[\mathbb{E}_{s \sim \rho}[\langle Q_s^{u^t}, \pi_s^t - \pi_s^{t+1} \rangle]] \geq \mathbb{E}[V^t(\rho)] - \mathbb{E}[V^{t+1}(\rho)] - \frac{2\epsilon}{1-\gamma},$$

*where the expectation $\mathbb{E}$ is taken w.r.t. the data sampled by RNAC, and $\epsilon = \epsilon_{stat} + \epsilon_{bias}$ is the same as that in Lemma 4.*

*Proof of Lemma 6.* Taking $\pi = \pi^t$ in the fundamental inequality Eq (18) gives

$$\eta^t \langle Q_s^{u^t}, \pi_s^t - \pi_s^{t+1} \rangle \leq -\mathrm{KL}(\pi_s^t, \pi_s^{t+1}) - \mathrm{KL}(\pi_s^{t+1}, \pi_s^t) \leq 0,$$

which implies $\pi^{t+1}$ is indeed improving $\pi^t$ along $Q^{u^t}$ direction. We then have

$$\langle Q_s^{u^t}, \pi_s^t - \pi_s^{t+1} \rangle \geq \frac{1}{1-\gamma} \mathbb{E}_{s' \sim d_s^{\pi^{t+1}, \kappa^{t+1}}}[\langle Q_{s'}^{u^t}, \pi_{s'}^t - \pi_{s'}^{t+1} \rangle]$$

$$= -\frac{1}{1-\gamma} \mathbb{E}_{s' \sim d_s^{\pi^{t+1}, \kappa^{t+1}}}[\langle Q_{s'}^{\pi^t}, \pi_{s'}^{t+1} - \pi_{s'}^t \rangle] + \frac{1}{1-\gamma} \mathbb{E}_{s' \sim d_s^{\pi^{t+1}, \kappa^{t+1}}}[\langle Q_{s'}^{u^t} - Q_{s'}^{\pi^t}, \pi_{s'}^t - \pi_{s'}^{t+1} \rangle]$$

$$= -\frac{1}{1-\gamma} \mathbb{E}_{s' \sim d_s^{\pi^{t+1}, \kappa^{t+1}}, a' \sim \pi_{s'}^{t+1}}[A^t(s', a')] + \frac{1}{1-\gamma} \mathbb{E}_{s' \sim d_s^{\pi^{t+1}, \kappa^{t+1}}}[\langle Q_{s'}^{u^t} - Q_{s'}^{\pi^t}, \pi_{s'}^t - \pi_{s'}^{t+1} \rangle]$$

$$\geq V^t(s) - V^{t+1}(s) + \frac{1}{1-\gamma} \mathbb{E}_{s' \sim d_s^{\pi^{t+1}, \kappa^{t+1}}}[\langle Q_{s'}^{u^t} - Q_{s'}^{\pi^t}, \pi_{s'}^t - \pi_{s'}^{t+1} \rangle],$$

where the second inequality is by Lemma 8. The proof of the lemma is then concluded by taking expectation $\mathbb{E}[\mathbb{E}_{s \sim \rho}[\cdot]]$ on both sides and applying Lemma 4. $\square$

**Assumption 6.** *For initial state distribution $\rho$, there exists $M$ that $\sup_{\kappa \in \mathcal{P}} \|\frac{d_\rho^{*,\kappa}}{\rho}\|_\infty \leq M < \infty$.*

**Theorem 8** (Formal statement of Theorem 1). *Under **RNAC-Linear Setting** and Assumption 6, RNAC with geometrically increasing step sizes $\eta^t \geq \frac{\frac{M}{1-\gamma}}{1 - \frac{1-\gamma}{M}} \eta^{t-1}$ for each $t = 1, 2, \ldots, T$, satisfies*

$$V^*(\rho) - \mathbb{E}[V^T(\rho)] \leq \left(1 - \frac{1-\gamma}{M}\right)^{T-1} \left((1 - \frac{1-\gamma}{M})(V^*(\rho) - V^0(\rho)) + \log|\mathcal{A}|\right)$$

$$+ \left(\frac{2}{1-\gamma} + \frac{2M}{(1-\gamma)^2}\right) \epsilon,$$

*where $\epsilon = \epsilon_{stat} + \epsilon_{bias}$ as in Lemma 4 with $\epsilon_{stat} = \tilde{O}(\frac{1}{\sqrt{K}} + \frac{1}{\sqrt{N}})$. Omitting the approximation $\epsilon_{bias}$, the sample complexity for achieving $\varepsilon$ robust optimal value (i.e., $V^*(\rho) - \mathbb{E}[V^T(\rho)] \leq \varepsilon$) is $\tilde{O}(1/\varepsilon^2)$ by taking $N = K = \tilde{\Theta}(1/\varepsilon^2)$ and $T = \Theta(\log(1/\varepsilon))$.*

*Proof of Theorem 8.* Taking $\pi = \pi^*$ in inequality (18), we have

$$\langle Q_s^{u^t}, \pi_s^* - \pi_s^t \rangle + \langle Q_s^{u^t}, \pi_s^t - \pi_s^{t+1} \rangle \leq \frac{1}{\eta^t} \mathrm{KL}(\pi_s^*, \pi_s^t) - \frac{1}{\eta^t} \mathrm{KL}(\pi_s^*, \pi_s^{t+1}). \tag{23}$$

We then take the expectation $\mathbb{E}[\mathbb{E}_{s\sim d_\rho^{*,\kappa t}}[\cdot]]$ on both sides. Note that

$$\mathbb{E}[\mathbb{E}_{s\sim d_\rho^{*,\kappa t}}[\langle Q_s^{u^t}, \pi_s^* - \pi_s^t\rangle]] = \mathbb{E}[\mathbb{E}_{s\sim d_\rho^{*,\kappa t}}[\langle Q_s^{\pi^t}, \pi_s^* - \pi_s^t\rangle]] + \mathbb{E}[\mathbb{E}_{s\sim d_\rho^{*,\kappa t}}[\langle Q_s^{u^t} - Q_s^{\pi^t}, \pi_s^* - \pi_s^t\rangle]]$$
$$\geq (1-\gamma)(V^*(\rho) - \mathbb{E}[V^t(\rho)]) - 2\epsilon,$$

where the inequality is by Lemma 8 and Lemma 4. Then by Lemma 6 and Assumption 6 that $M = \sup_\kappa \|\frac{d_\rho^{*,\kappa}}{\rho}\|_\infty$, we have

$$\mathbb{E}[\mathbb{E}_{s\sim d_\rho^{*,\kappa t}}[\langle Q_s^{u^t}, \pi_s^t - \pi_s^{t+1}\rangle]] \geq M\left(\mathbb{E}[V^t(\rho)] - \mathbb{E}[V^{t+1}(\rho)] - \frac{2\epsilon}{1-\gamma}\right),$$

since $\langle Q_s^{u^t}, \pi_s^t - \pi_s^{t+1}\rangle \leq 0$. The outside expectation $\mathbb{E}$ is taken w.r.t. the data sampled by RNAC, and since all the following statements are under expectation $\mathbb{E}$, we omit $\mathbb{E}$ in the proof for simplicity and bring it back at the end. Combining the inequality above and Eq (23), we have

$$(1-\gamma)(V^*(\rho) - V^t(\rho)) - 2\epsilon + M\left(V^t(\rho) - V^{t+1}(\rho) - \frac{2\epsilon}{1-\gamma}\right)$$
$$\leq \frac{1}{\eta^t}\mathrm{KL}_{d_\rho^{*,\kappa t}}(\pi^*, \pi^t) - \frac{1}{\eta^t}\mathrm{KL}_{d_\rho^{*,\kappa t}}(\pi^*, \pi^{t+1}).$$

It follows that

$$V^*(\rho) - V^{t+1}(\rho) \leq \left(1 - \frac{1-\gamma}{M}\right)(V^*(\rho) - V^t(\rho)) + \frac{1}{M\eta^t}\mathrm{KL}_{d_\rho^{*,\kappa t}}(\pi^*, \pi^t)$$
$$- \frac{1}{M\eta^t}\mathrm{KL}_{d_\rho^{*,\kappa t}}(\pi^*, \pi^{t+1}) + 2\left(\frac{\epsilon}{M} + \frac{\epsilon}{1-\gamma}\right),$$

which implies

$$V^*(\rho) - V^T(\rho) \leq \left(1 - \frac{1-\gamma}{M}\right)^T(V^*(\rho) - V^0(\rho)) + \left(1 - \frac{1-\gamma}{M}\right)^{T-1}\frac{\mathrm{KL}_{d_\rho^{*\kappa 0}}(\pi^*, \pi^0)}{M\eta^0}$$
$$+ \sum_{t=1}^{T-1}\left(1 - \frac{1-\gamma}{M}\right)^{T-1-t}\left(\frac{1}{M\eta^t}\mathrm{KL}_{d_\rho^{*,\kappa t}}(\pi^*, \pi^t) - \frac{1 - \frac{1-\gamma}{M}}{M\eta^{t-1}}\mathrm{KL}_{d_\rho^{*,\kappa t-1}}(\pi^*, \pi^t)\right)$$
$$+ \frac{2M}{1-\gamma}\left(\frac{\epsilon}{M} + \frac{\epsilon}{1-\gamma}\right).$$

Take geometrically increasing step sizes $\eta^t \geq \frac{\frac{M}{1-\gamma}}{1 - \frac{1-\gamma}{M}}\eta^{t-1}$ for each $t$. Since $M = \sup_\kappa \|\frac{d_\rho^{*,\kappa}}{\rho}\|_\infty \geq (1-\gamma)\sup_{\kappa,\kappa'}\|\frac{d_\rho^{*,\kappa}}{d_\rho^{*,\kappa'}}\|_\infty$, we have

$$\frac{1}{M\eta^t}\mathrm{KL}_{d_\rho^{*,\kappa t}}(\pi^*, \pi^t) - \frac{1 - \frac{1-\gamma}{M}}{M\eta^{t-1}}\mathrm{KL}_{d_\rho^{*,\kappa t-1}}(\pi^*, \pi^t)$$
$$\leq \frac{1 - \frac{1-\gamma}{M}}{M\eta^{t-1}}\left(\frac{1-\gamma}{M}\mathrm{KL}_{d_\rho^{*,\kappa t}}(\pi^*, \pi^t) - \mathrm{KL}_{d_\rho^{*,\kappa t-1}}(\pi^*, \pi^t)\right) \leq 0.$$

We bring back the expectation $\mathbb{E}$ over the data and finally arrive at

$$V^*(\rho) - \mathbb{E}[V^T(\rho)] \leq \left(1 - \frac{1-\gamma}{M}\right)^T(V^*(\rho) - V^0(\rho)) + \left(1 - \frac{1-\gamma}{M}\right)^{T-1}\frac{\mathrm{KL}_{d_\rho^{*\kappa 0}}(\pi^*, \pi^0)}{M\eta^0}$$
$$+ \frac{2M}{1-\gamma}\left(\frac{\epsilon}{M} + \frac{\epsilon}{1-\gamma}\right),$$

where $\mathrm{KL}_{d_\rho^{*\kappa 0}}(\pi^*, \pi^0) \leq \log|\mathcal{A}|$ when $\pi^0$ is a uniform policy. $\qquad\square$

**Theorem 9** (Formal statement of Theorem 2). *Under **RNAC-Linear Setting** and Assumption 6 when IPM uncertainty set is considered, take $\rho = \nu^{\pi^*}$ and $\eta^t = \eta = (1-\gamma)\log(|\mathcal{A}|)$, RNAC satisfies*

$$V^*(\rho) - \frac{1}{T}\sum_{t=1}^T\mathbb{E}[V^t(\rho)] \leq \frac{2}{(1-\gamma)(1-\beta)T} + \frac{2}{1-\beta}\frac{2-\gamma}{1-\gamma}\epsilon,$$

where $\beta$ is in Assumption 6 (guaranteed by Proposition 4 for DS uncertainty set and assumed for IPM uncertainty set), and $\epsilon = \epsilon_{stat} + \epsilon_{bias}$ as in Lemma 4 with $\epsilon_{stat} = \tilde{O}(\frac{1}{\sqrt{K}} + \frac{1}{\sqrt{N}})$. Omitting the approximation $\epsilon_{bias}$, the sample complexity for achieving $\varepsilon$ robust optimal value on average (i.e., $V^*(\rho) - \frac{1}{T}\sum_{t=1}^{T}\mathbb{E}[V^t(\rho)] \leq \varepsilon$) is $\tilde{O}(1/\varepsilon^3)$ by $N = K = \tilde{\Theta}(1/\varepsilon^2)$ and $T = \Theta(1/\varepsilon)$.

*Proof of Theorem 9.* Taking $\pi = \pi^*$ in inequality (18), we have

$$\langle Q_s^{u^t}, \pi_s^* - \pi_s^t \rangle + \langle Q_s^{u^t}, \pi_s^t - \pi_s^{t+1} \rangle \leq \frac{1}{\eta}\mathrm{KL}(\pi_s^*, \pi_s^t) - \frac{1}{\eta}\mathrm{KL}(\pi_s^*, \pi_s^{t+1}).$$

Since all the following statements are under expectation $\mathbb{E}$ over the data, we omit $\mathbb{E}$ in the proof for simplicity and bring it back at the end. According to Lemma 9 and Lemma 4, we have

$$\mathbb{E}_{s\sim\rho}[\langle Q_s^{u^t}, \pi_s^* - \pi_s^t \rangle] = \mathbb{E}_{s\sim\rho}[\langle Q_s^{\pi^t}, \pi_s^* - \pi_s^t \rangle] + \mathbb{E}_{s\sim\rho}[\langle Q_s^{u^t} - Q_s^{\pi^t}, \pi_s^* - \pi_s^t \rangle]$$
$$\geq (1 - \beta)(V^*(\rho) - V^t(\rho)) - 2\epsilon.$$

Based on Lemma 6, we have

$$\mathbb{E}_{s\sim\rho}[\langle Q_s^{u^t}, \pi_s^t - \pi_s^{t+1} \rangle] \geq V^t(\rho) - V^{t+1}(\rho) - \frac{2\epsilon}{1-\gamma}.$$

It then follows that

$$(1-\beta)(V^*(\rho) - V^{t+1}(\rho)) \leq V^{t+1}(\rho) - V^t(\rho) + \frac{\mathrm{KL}_\rho(\pi^*, \pi^t) - \mathrm{KL}_\rho(\pi^*, \pi^{t+1})}{\eta} + \frac{2-\gamma}{1-\gamma}2\epsilon.$$

Taking summation from $t = 0$ to $T - 1$ from both sides we have

$$V^*(\rho) - \frac{1}{T}\sum_{t=1}^{T}V^t(\rho) \leq \frac{V^T(\rho)}{(1-\beta)T} + \frac{\mathrm{KL}_\rho(\pi^*, \pi^0)}{\eta(1-\beta)T} + \frac{2\epsilon}{1-\beta}\frac{2-\gamma}{1-\gamma}.$$

We then conclude the theorem by choosing $\eta = (1 - \gamma)\log(|\mathcal{A}|)$ and bringing back the expectation $\mathbb{E}$ over data,

$$V^*(\rho) - \frac{1}{T}\sum_{t=1}^{T}\mathbb{E}[V^t(\rho)] \leq \frac{2}{(1-\gamma)(1-\beta)T} + \frac{2}{1-\beta}\frac{2-\gamma}{1-\gamma}\epsilon.$$

$\square$

**Discussion:** The linear convergence for NPG with linear function approximation has been previously studied in canonical MDP [4]. We present the linear convergence of RNAC (Algorithm 1) in Theorem 8. Compared to Theorem 8, Theorem 9 utilizing constant step sizes and leads to sublinear convergence, and it is proved only for initial state distribution $\rho = \nu^{\pi^*}$. Moreover, Theorem 9 does not require Assumption 6 though may need Assumption 2 for IPM uncertainty set.

### E.2 General Function Approximation

**RNAC-General Setting.** We study the RNAC (Algorithm 1) with robust critic performing FRVE (Algorithm 5) and robust natural actor performing RNPG (Algorithm 6 with $\zeta_n = \Theta(1/n)$ as in Theorem 7) under general value function approximation as in Eq (14) and general policy class as in Eq (20), for DS uncertainty set taking $\delta$ suggested by Proposition 4. We assume Assumption 1, Assumption 5 and Assumption 4 ($\|u_*^t\| \leq U$), hold.

**Theorem 10.** *Under **RNAC-General Setting**, suppose $\log \pi_\theta(a|s)$ is an $L$-smooth function of $\theta$. Take $\rho = \nu^{\pi^*}$ and $\eta^t = \eta = \sqrt{\frac{2\mathrm{KL}_\rho(\pi^*, \pi^0)}{LU^2 T}}$ for each $t$, RNAC satisfies*

$$V^*(\rho) - \mathbb{E}\left[\frac{1}{T}\sum_{t=0}^{T-1}V^t(\rho)\right] \leq \frac{\sqrt{2LU^2\mathrm{KL}_\rho(\pi^*, \pi^0)}}{(1-\beta)\sqrt{T}} + \frac{\epsilon'}{1-\beta},$$

*where $\beta$ is in Assumption 6 (guaranteed by Proposition 4 for DS uncertainty set), and $\epsilon' = \epsilon'_{stat} + \epsilon'_{bias}$ as in Lemma 5 with $\epsilon'_{stat} = \tilde{O}(\frac{1}{\sqrt{K}} + \frac{1}{\sqrt{N}})$. Omitting the approximation $\epsilon'_{bias}$, the sample complexity for achieving $\varepsilon$ robust optimal value is $\tilde{O}(1/\varepsilon^4)$ by $N = K = \tilde{\Theta}(1/\varepsilon^2)$ and $T = \Theta(1/\varepsilon^2)$.*

*Proof.* We have

$$(1-\beta)(V^*(\rho) - \mathbb{E}[V^t(\rho)]) \leq \mathbb{E}[\mathbb{E}_{(s,a)\sim\rho\circ\pi^*}[A^t(s,a)]]$$

$$= \mathbb{E}[\mathbb{E}_{(s,a)\sim\rho\circ\pi^*}[\nabla_\theta \log \pi^t(a|s)^\top u^t] + \mathbb{E}[\mathbb{E}_{(s,a)\sim\rho\circ\pi^*}[A^t(s,a) - A^{u^t}(s,a)]]$$

$$\leq \mathbb{E}\left[\mathbb{E}_{(s,a)\sim\rho\circ\pi^*}\left[\frac{1}{\eta}\log\frac{\pi^{t+1}(a|s)}{\pi^t(a|s)}\right]\right] + \frac{\eta L\mathbb{E}[\|u^t\|^2]}{2} + \epsilon'$$

$$\leq \frac{1}{\eta}\mathbb{E}[\mathrm{KL}_\rho(\pi^*,\pi^t) - \mathrm{KL}_\rho(\pi^*,\pi^{t+1})] + \frac{\eta LU^2}{2} + \epsilon',$$

where the first inequality is by Lemma 9, the first equality is by the definition of $A^{u^t}(s,a) = \nabla\log\pi^t(a|s)^\top u^t$, and the second inequality is by the $L$-smoothness of $\log\pi_\theta(a|s)$ and Lemma 5.

We then conclude the proof by summing from $t = 0$ to $T-1$ from both sides and taking $\eta = \sqrt{\frac{2\mathrm{KL}_\rho(\pi^*,\pi^0)}{LU^2T}}$. $\qquad\square$

Note that in **RNAC-General Setting**, the critic is employing general function approximation, where the IPM uncertainty set is not defined (c.f. Section C.2 for more discussion). It is not hard to show a similar result as in Theorem 10 for the RNAC with robust critic performing RLTD (Algorithm 2) and robust natural actor performing RNPG (Algorithm 6) for both DS and IPM uncertainty sets.

## F  Supporting Lemmas

We introduce some supporting lemmas for proving the results of this paper.

### F.1  Performance Difference Lemmas

A key supporting lemma in the proof of convergence of policy gradient-based methods is the performance difference lemma. In the canonical RL with a fixed transition kernel $\kappa$, we have the following performance lemma.

**Lemma 7** (Performance difference [23]). *For any policy $\pi, \pi'$, and transition $\kappa$, we have*

$$V_\kappa^{\pi'}(\rho) - V_\kappa^\pi(\rho) = \frac{1}{1-\gamma}\mathbb{E}_{s\sim d_\rho^{\pi',\kappa}}\mathbb{E}_{a\sim\pi'(\cdot|s)}[A^\pi(s,a)].$$

However, due to the non-singleton uncertainty set $\mathcal{P}$, the worst-case transition kernel is a function of policy $\pi$, and we have the following performance difference inequality lemma.

**Lemma 8** (Robust performance difference). *For any policy $\pi, \pi'$, denote $\kappa' = \kappa_{\pi'}$ and $\kappa = \kappa_\pi$ be their worst-case transition kernels, respectively. Then*

$$\frac{1}{1-\gamma}\mathbb{E}_{s\sim d_\rho^{\pi',\kappa'}}\mathbb{E}_{a\sim\pi'(\cdot|s)}[A^\pi(s,a)] \leq V^{\pi'}(\rho) - V^\pi(\rho) \leq \frac{1}{1-\gamma}\mathbb{E}_{s\sim d_\rho^{\pi',\kappa}}\mathbb{E}_{a\sim\pi'(\cdot|s)}[A^\pi(s,a)].$$

*Proof of Lemma 8.* The LHS is by

$$V_{\kappa'}^{\pi'}(\rho) - V^\pi(\rho) = \mathbb{E}_{\kappa',\pi'}\left[\sum_{t\geq 0}\gamma^t(r(s_t,a_t) + V^\pi(s_t) - V^\pi(s_t))\right] - V^\pi(\rho)$$

$$= \mathbb{E}_{\kappa',\pi'}\left[\sum_{t\geq 0}\gamma^t(r(s_t,a_t) + \gamma V^\pi(s_{t+1}) - V^\pi(s_t))\right]$$

$$\geq \frac{1}{1-\gamma}\mathbb{E}_{s\sim d_\rho^{\pi',\kappa'}}\mathbb{E}_{a\sim\pi'(\cdot|s)}[A^\pi(s,a)].$$

The RHS is by

$$V_{\kappa'}^{\pi'}(\rho) - V_\kappa^\pi(\rho) \leq V_\kappa^{\pi'}(\rho) - V_\kappa^\pi(\rho) = \frac{1}{1-\gamma}\mathbb{E}_{s\sim d_\rho^{\pi',\kappa}}\mathbb{E}_{a\sim\pi'(\cdot|s)}[A^\pi(s,a)].$$

$\qquad\square$

Moreover, the optimality gap between the optimal policy $\pi^*$ and any policy $\pi$ is upper bounded as in the lemma below.

**Lemma 9** (Robust optimality gap lemma). *Under Assumption 2, for any $\pi$ and any $\rho$,*

$$V^*(\rho) - V^\pi(\rho) \leq \frac{1}{1-\beta} \mathbb{E}_{s \sim d_\rho^{\pi^*, p^\circ}, a \sim \pi_s^*}[A^\pi(s, a)].$$

*Proof of Lemma 9.*

$$V^*(\rho) - V^\pi(\rho) = \mathbb{E}_{s \sim \rho} \mathbb{E}_{a \sim \pi_s^*}[r(s, a) + \gamma \sum_{s'} \kappa_{\pi^*}(s'|s, a)V^*(s')] - V^\pi(\rho)$$

$$= \mathbb{E}_{s \sim \rho} \mathbb{E}_{a \sim \pi^*}[r(s, a) + \gamma \sum_{s'} \kappa_\pi(s'|s, a)V^\pi(s') - V^\pi(\rho)]$$

$$+ \gamma \mathbb{E}_{s \sim \rho} \mathbb{E}_{a \sim \pi^*} \sum_{s'} (\kappa_{\pi^*}(s'|s, a)V^*(s') - \kappa_{\pi^t}(s'|s, a)V^\pi(s'))$$

$$\leq \mathbb{E}_{s \sim \rho} \mathbb{E}_{a \sim \pi^*}[A^\pi(s, a)] + \gamma \mathbb{E}_{s \sim \rho} \mathbb{E}_{a \sim \pi^*} \sum_{s'} \kappa_{\pi^t}(s'|s, a)(V^*(s') - V^\pi(s'))$$

$$\leq \mathbb{E}_{s \sim \rho} \mathbb{E}_{a \sim \pi^*}[A^\pi(s, a)] + \beta \mathbb{E}_{s \sim \rho} \mathbb{E}_{a \sim \pi^*} \sum_{s'} p_{s,a}^\circ(s')(V^*(s') - V^\pi(s'))$$

$$\leq \cdots$$

$$\leq \frac{1}{1-\beta} \mathbb{E}_{s \sim d_\rho^{\pi^*, p^\circ}, a \sim \pi_s^*}[A^\pi(s, a)].$$

$\square$

### F.2 Concentration Lemmas with Markov samples

**Assumption 7** (Uniformly geometric mixing, quantitative restatement of Assumption 1). *There exists some $0 < \lambda < 1$ and $C > 0$ such that for any policy $\pi$, the Markov chain $\{s_k\}$ induced by applying $\pi$ in the nominal model $p^\circ$ is geometrically ergodic with unique stationary distribution $\nu^\pi$, i.e., $\max_s \|\mathbb{P}(s_k \in \cdot | s_0 = s) - \nu^\pi\|_{TV} \leq C\lambda^k$, $\forall k$. The uniform convergence to unique stationary distribution implies that for any policy $\pi$ the spectral gap (i.e., the difference between the largest and the second largest eigenvalues) of the state transition kernel has a strictly positive lower bound.*

**Lemma 10** (Bernstein's inequality for stationary Markovian data, Theorem 2 in [21]). *Suppose $Z_1, Z_2, \ldots, Z_n$ follow some stationary Markovian data in space $\mathcal{Z}$ with stationary distribution $\nu \in \Delta_{\mathcal{Z}}$ and spectral gap of the Markov chain strictly greater than 0. For any function $f : \mathcal{Z} \to [-c, c]$ that, $\mathbb{E}_{Z \sim \nu}[f(Z)] = 0$ and $\mathbb{E}_{Z \sim \nu}[f(Z)^2] = \sigma^2$, with probability at least $1 - \delta$,*

$$\frac{1}{n} \sum_{i=1}^n f(Z_i) \leq C_1 \sqrt{\frac{\sigma^2 \ln(1/\delta)}{n}} + C_2 \frac{c \ln(1/\delta)}{n},$$

*for some spectral gap-dependent multiplicative factors $C_1, C_2$.*

Since we assume Assumption 7 throughout the paper during the analysis, the spectral gap is uniformly lower bounded by some constant great than zero, and we view $C_1, C_2$ as some constant that can be omitted in the big-O notation. We refer to [21] for further details of Bernstein's inequality for stationary Markovian data.

**Lemma 11** (Lemma A.11 in [2] with Markovian data). *Consider a stationary Markov chain on domain $\mathcal{X} \times \mathcal{Z}$ that is geometrically mixing with transition $p(x', z'|x, z) = p_{X'|Z}(x'|z)p_{Z|X}(z'|x)$ and stationary distribution $\nu \circ p_{Z|X}$. Let $\mathcal{D} = \{(X_1, Z_1), (X_2, Z_2), \ldots, (X_n, Z_n)\}$ be stationary Markovian data sampled with $X_1 \sim \nu$. Given a function class $\mathcal{G} \subset \mathbb{R}^{\mathcal{X}}$ and a function $G : \mathcal{X} \times \mathcal{Z} \to \mathbb{R}$ that $\mathbb{E}_{Z \sim p_{Z|X}(\cdot|x)}[G(x, Z)] = g^*(x)$ and $|G(x, z)| \leq G_{max}$. Let $\hat{g} = \arg\min_{g \in \mathcal{G}} \frac{1}{2} \sum_{(x,z) \in \mathcal{D}} (G(x, z) - g(x))^2$ be the MSE estimate of $g^*$ within $\mathcal{G}$, then with probability at least $1 - \delta$,*

$$\sqrt{\mathbb{E}_{x \sim \nu}\left[(\hat{g}(x) - g^*(x))^2\right]} \leq O\left(G_{max}\sqrt{\frac{\log(|\mathcal{G}|/\delta)}{n}}\right) + O(\epsilon_{bias}), \tag{24}$$

*where* $\epsilon_{bias} = \min_{g \in \mathcal{G}} \|g - g^*\|_{\nu}$.

*Proof.* The proof follows the same as [2]. The only difference is replacing the canonical Bernstein's inequality with i.i.d. data by Bernstein's inequality for stationary Markovian data in Lemma 10. $\square$

### F.3 Convergence of Stochastic Approximation

$F : \mathbb{R}^d \times \mathcal{Z} \to \mathbb{R}^d$ is a stochastic operator. The stochastic approximation algorithm updates the parameter by $w_{k+1} = w_k + \alpha_k F(w, Z_k)$, where $(Z_0, Z_1, \ldots)$ is a Markov chain, that is geometrically ergodic for some $C > 0, 1 > \lambda > 0$ with stationary distribution $\nu$, i.e., $\max_z \|\mathbb{P}(Z_k \in \cdot | Z_0 = z) - \nu\|_{TV} \leq C\lambda^k$, $\forall k$. (c.f. Assumption 1). The stochastic approximation algorithm solves the following equation

$$0 = \bar{F}(w) := \mathbb{E}_{Z \sim \nu}[F(w, Z)],$$

under the following assumption and appropriate step sizes $(\alpha_1, \alpha_2, \ldots)$.

**Assumption 8.** *1.* $(Z_0, Z_1, \ldots)$ *are geometrically mixed to the stationary distribution.*

2. $\exists L_1 > 0$ *that* $\|F(w, z) - F(w', z)\| \leq L_1\|w - w'\|$, $\|F(0, z)\| \leq L_1$ *for any* $w, w' \in \mathbb{R}^d, z \in \mathcal{Z}$.

3. $\bar{F}(w) = 0$ *has unique solution* $w^*$ *and there exists* $c_0 > 0$ *that* $(w - w^*)^\top \bar{F}(w) \leq -c_0\|w - w^*\|^2$ *for any* $w \in \mathbb{R}^d$.

The second condition of Assumption 8 is a Lipschitz condition of the operator $F$, and the third condition is the "strongly-concavity" structure of the averaged operator $\bar{F}$.

**Lemma 12** (Big-O version of Corollary 2.1.2 in [10])**.** *Under Assumption 8, taking appropriate* $\alpha_k = \Theta(1/k)$ *gives* $\mathbb{E}[\|w_K - w^*\|^2] = \tilde{O}(1/K)$.

To apply the lemma above, we need to show that the proposed operator $F$ satisfies Assumption 8.

## G   More Related Works

The framework of the robust Markov decision process (RMDP) is proposed by [19, 40] to learn the optimal robust policy that achieves the optimal worst-case performance over all possible models in the uncertainty set. For the planning problem of RMDP, Iyengar [19] and Nilim et al. [40] show that value iteration achieves linear convergence to the optimal robust values.

When the transition model is unknown, several model-based and value-based approaches have been proposed and studied for robust RL in both the tabular setting and the function approximation setting. Under the tabular setting, Xu et al. [64] design the model-based robust phase value learning algorithm and demonstrates an $O(1/\epsilon^2)$ sample complexity bound. Under the function approximation setting, Panaganti et al. [43] develop a robust fitted Q-iteration with the total variation uncertainty set and show an $O(1/\epsilon^2)$ sample complexity to achieve approximate optimality and [6] develop a model-based method with the same sample complexity under certain coverage assumption. However, these methods are not scalable to continuous control problems. Instead, policy-based approaches directly parameterize policy and have more representation power in modeling stochastic policies, more flexibility for policy manipulation, and a better ability to solve robotics control problems with large action space.

The existing policy-based methods with convergence guarantees mainly focus on the tabular setting. Wang et al. [61] show an $O(1/\epsilon^7)$ sample complexity for robust actor-critic with $R$-contamination uncertainty sets. Li et al. [29] prove an $\tilde{O}(1/\epsilon^2)$ sample complexity for robust natural actor-critic assuming the existence of an oracle to solve the inner optimization problem. Kumar et al. [27] develop a robust policy gradient method based on $\ell_p$ norm uncertainty sets and extends its formulation under the $s$-rectangular assumption. All works mentioned above are limited to the tabular case with critic Q and adopt computationally infeasible uncertainty sets in the case of the large state space. Kuang et al. [26] illustrate a state disturbance view of Wasserstein metric-based RMDP, which can be generalized to the large state space. However, they only provide a policy iteration guarantee in the tabular setting (contraction under $\| \cdot \|_{\infty}$). Instead, we focus on large-scale robust RL, propose

two computationally efficient uncertainty sets, and demonstrate an $\tilde{O}(1/\epsilon^2)$ (resp., $\tilde{O}(1/\epsilon^4)$) sample complexity under linear (resp., general) function approximation. Additionally, we regard $V$ as a critic instead of Q, which is closer to the actual implementation of on-policy natural actor-critic algorithms.

