# OpenReview forum: "Natural Actor-Critic for Robust Reinforcement Learning with Function Approximation"
_NeurIPS.cc/2023/Conference — NeurIPS 2023 poster_

### Official Review · Reviewer_4Et9 · 2023-07-04

**Soundness:** 3 good
**Presentation:** 3 good
**Contribution:** 2 fair
**Rating:** 5
**Confidence:** 4

**Summary:**

This paper studies the actor-critic approach for robust RL. Especially, a Double-Sampling Uncertainty Set and an Integral Probability Metric Uncertainty Set are developed to overcome the curse of problem scale. A robust natural Actor-Critic algorithm is then proposed with convergence results. A significant number of experiments are designed.

**Strengths:**

The paper is well-written and clear in general. The design of the two uncertainty sets is new and novel, which shows advantages under large-scale problems.

**Weaknesses:**

1. The motivation for designing the two uncertainty sets is somehow unclear to me. I understand there is uncertainty in the sets designed, but don't understand the motivation of this uncertainty set. In lines 640-646, the authors explain the uncertainty set contains transition kernels that are perturbed from the uniform distribution, this explanation seems unclear to me, and I can't understand the motivation for such a definition.
2. One of the critical problems in studying robust RL with function approximation is the contraction of the approximated Bellman operator. The approach used in this paper is similar to the previous ones, i.e., use conditions on the radius of the uncertainty set and discount factor. This hence reduces the novelty and contribution of the paper.

**Questions:**

Can you explain the motivation for the designing of the uncertainty sets? In what sense do they imply robustness and uncertainty?

**Limitations:**

See the parts above.

---

> ### Author Rebuttal · Authors · 2023-08-09
>
> We are encouraged by the reviewer's comments that the paper is well-written, and that the designed uncertainty sets are novel and show advantages for large-scale problems. Below, we give a detailed response to your comments. We believe that we have addressed all your concerns, and we sincerely hope that the reviewer would consider increasing their score. Please note that line numbers are based on the supplementary material (RNAC-full.pdf).
>
> **Q1.** "The motivation for designing the two uncertainty sets is somehow unclear..."
>
> **Response:** From the perspective of the development of robust RL in the literature, previous works have considered multiple types of uncertainty sets, such as KL uncertainty set, Total-Variation uncertainty set, and Chi-square uncertainty set. However, as we have discussed in  Lines 42-49 and Appendix B, these uncertainty sets do not scale up for function approximation in large state spaces. This motivates the design of tractable uncertainty sets for large state spaces. The tractability is provided by the computationally efficient empirical robust Bellman operators of the designed uncertainty sets (Eq (3) and Eq (6)).
>
> The Double-Sampling (DS) uncertainty set (cf. Lines 134-140 and 640-646) can be motivated and explained as follows. In canonical RL, the next-step state $s'$ sampled from the nominal model, i.e., $s' \sim p^0_{s, a}$ can be viewed equivalently as a double-sampling process: first generate $m$ states  $s_1', s_2', \ldots, s_m'$ sampled i.i.d. according to $p^0_{s, a}$ , and then uniformly select one from these $m$ states as the next-step state $s'$. Double sampling gets its name from these two phases of sampling. With this interpretation, we can thus perturb the nominal transition $p^0_{s, a}$ by perturbing the second phase sampling from uniform selection. We let the selection distribution be $\alpha \in \Delta(m)$ and allow it to deviate from the uniform distribution $\text{Unif}(m)$ as $\mathrm{d}(\alpha, \text{Unif}(m)) \leq \delta$. Due to this perturbation of the second phase of the double sampling process ($\alpha$ can depend on the generated $m$ states of the first phase), the induced next-step state $s'$ will not follow the nominal kernel $p^0_{s, a}$ but a new kernel that lies in the uncertainty set. The DS uncertainty set is implicitly defined as the set of all such new kernels, which is determined by $m, \mathrm{d}(\cdot, \cdot), \delta$.
>
> IPM is a general class of divergence measures that contains many metrics as special cases for different function classes (cf. Lines 158-160). IPM takes advantage of the prior information in the function class, which is helpful when considering RL with a large state space under function approximation. We take a function class based on the feature matrix Eq. (5), and the designed IPM-based uncertainty set based on this function class takes advantage of the underlining structure of the value function approximation.
>
> Please let us know if the above discussion is clearer for motivating the design of the  uncertainty sets. If so, we will add the discussion in the revision.
>
> **Q2.** "...The approach used in this paper is similar to the previous ones, i.e., use conditions on the radius of the uncertainty set and discount factor. This hence reduces the novelty and contribution of the paper."
>
> **Response:**
> While we acknowledge the partial adoption of the approach in [47] by Assumption 2 for the DS uncertainty set (Proposition 2), it is essential to emphasize that [47] does not offer any specific uncertainty set satisfying Assumption 2. Our paper introduces technical novelty by demonstrating that the commonly considered f-divergence may violate Assumption 2 (Proposition 3). Additionally, for the IPM uncertainty sets, their contraction property shown in Lemma 1 is independent of the previous approach. Furthermore, a large volume of the analysis in the paper is regarding the convergence of the proposed algorithm (e.g., Theorems 1-4 in the main paper), which also constitutes a significant technical contribution. The fact that the proof of Proposition 2 partly leverages a previous approach diminishes neither the overall novelty and contribution of the paper, nor the technical contribution.
>
> In addition, the major novelty of the paper can be attributed to the design of the new uncertainty sets and the robust policy-based algorithm. The major contributions, besides the novel uncertainty sets and algorithms, are both theoretical -- convergence guarantees under function approximation for robust RL, as well as empirical -- extensive experiments in Mujoco simulation and real-world TurtleBots suggesting robust behavior of the proposed algorithm. By providing guarantees we have successfully closed an important gap in the literature on a policy-based approach for robust RL under function approximation.
>
> [47] Aviv Tamar, Shie Mannor, and Huan Xu. Scaling up robust mdps using function approximation. In International conference on machine learning, pages 181–189. PMLR, 2014.
>
> **Q3.** "In what sense do they imply robustness and uncertainty?"
>
> **Response:** By the definition of the uncertainty set for the Robust MDP (RMDP, cf. Line 103-119), the results establish robustness in the sense that the optimal policy of the RMDP achieves the best performance for the worst transition model in the uncertainty set. Equipped with the newly designed DS or IPM uncertainty sets, the proposed algorithm aims to find the optimal policy of the corresponding RMDP. We have demonstrated that the proposed algorithm with either DS or IPM uncertainty sets results in robust policy, in the sense that the learned policies have more stable performances in perturbed environments, as shown in Figures 1-2 in Section 7, Figures 4-5 in Appendix A, and Figures 1-2 in the attached new pdf.

---

> > ### Comment · Reviewer_4Et9 · 2023-08-21
> >
> > The response solves my concerns, and I hence increase my score.

---

> > > ### Author Response · Authors · 2023-08-21
> > >
> > > We thank the reviewer for carefully reading our rebuttal and raising the score. We will add the above clarification in our final version.

---

### Official Review · Reviewer_r1LZ · 2023-07-05

**Soundness:** 4 excellent
**Presentation:** 2 fair
**Contribution:** 3 good
**Rating:** 7
**Confidence:** 3

**Summary:**

This paper tackles robust reinforcement learning in large state spaces, where the transition kernel is accessible only in a nominal setting. The authors demonstrate that the $f$-divergence, R-contamination, and the $l_{p}$ norm are computationally infeasible in the context of robust RL for large state spaces. To overcome this limitation, the authors propose two new tractable uncertainty set formulations suitable for large dimensions: double sampling (DS) and the integral probability metric (IPM).
DS involves independently and identically distributed state sampling following a transition kernel $p^o_{s,a}$. The IPM corresponds to the robust Bellman operator, but with a regularization term on the norm of weights. Both approaches are enabling Robust RL in scenarios previously hindered by computational complexity.
The paper introduces a new algorithm, the Robust Natural Actor Critic (RNAC), for training both a critic and an actor for the proposed IPM and double sampling uncertainty sets. The authors provide convergence guarantees for the RNAC algorithm.
Finally, the authors demonstrate the efficency of their approach via two applications: one involving the suite of MuJoCo environments and the other, a real-world robotics application. These practical applications lend credence to the theoretical contributions and the robustness of the proposed approach.


**Strengths:**

- The authors provide two straightforward and computationally feasible uncertainty set formulations for large state spaces.
- The paper offers substantial theoretical contributions.
- The real-world robotics application lends credibility to the paper's robustness claims.


**Weaknesses:**

- The convergence guarantees are valid for $(s,a)$-rectangular uncertainty sets, which is rather limiting. However, it is commendable that the authors have acknowledged this limitation in their work and suggested it as an avenue for future research.
- The paper seems incomplete without the appendix. The need to constantly refer to the appendix disrupts the flow of reading.


**Questions:**

- In lines 175-177, why did you not apply regularization to the bias parameter in the last layer? Moreover, this seems inconsistent with the provided implementation (lines 201-206 in the `RNAC-ppo/ppo_continious.py` file), where regularization is applied to all the critic's layers.

- Doesn't the IPM's proposed regularization ultimately reduce the Lipschitz constant of the learned function, making the agent less sensitive to state variations?


**Limitations:**

The authors mention their limitations as future research directions. We appreciate the authors for being upfront.

---

> ### Author Rebuttal · Authors · 2023-08-09
>
> We thank the reviewer for the clear summary of the paper, finding our theoretical results of substantial contribution and our empirical evaluations credible.
>
> **Q1.** "The convergence guarantees are valid for $(s,a)$-rectangular uncertainty sets, which is rather limiting. However, it is commendable that the authors have acknowledged this limitation in their work and suggested it as an avenue for future research."
>
> **Response:** As the reviewer noted, the $(s,a)$-rectangularity assumption is now a standard assumption used in the robust MDP/RL literature that can yield  tractable theoretical analysis. We completely agree with the reviewer  that this assumption is indeed a limitation of the existing literature, including ours.  We believe alleviating this limitation while maintaining theoretical tractability is an essential endeavor for future research in robust RL.
>
> **Q2.** "The paper seems incomplete without the appendix. The need to constantly refer to the appendix disrupts the flow of reading."
>
> **Response:** We thank the reviewer's valuable feedback. We refer to the appendix for detailed or technical discussions in several places in the main paper due to space limits. To eliminate the disruptions of the flow, we will not refer to the appendix at each different place in the main paper, but we will simply add one remark discussing the appendix and its supporting relation to the sections all in one place. We would be glad to take action on any further suggestions from the reviewer to make our presentation better.
>
> **Q3.** "In lines 175-177, why did you not apply regularization to the bias parameter in the last layer? Moreover, this seems inconsistent with the provided implementation (lines 201-206 in the RNAC-ppo/ppo\_continious.py file), where regularization is applied to all the critic's layers."
>
> **Response:** It is theoretically proved in Proposition 1 that under the designed IPM uncertainty set, the regularization does not include the bias parameter. The regularization without bias parameter is thus an implication of the structure of the designed IPM uncertainty set.
> We thank the reviewer's careful examination of our code. The theory is consistent with our implementation, where in lines 201-206 of RNAC-ppo/ppo\_continous.py file, the last layer's bias term is not included in the regularization, cf. bias\_norm[0:-1] in line 206.
>
> **Q4.** "Doesn't the IPM's proposed regularization ultimately reduce the Lipschitz constant of the learned function, making the agent less sensitive to state variations?"
>
> **Response:** We agree with the reviewer's intuition that this regularization has the effect of reducing the Lipschitz constant of the value function, and thus the learned value function is less sensitive with respect to the state variations. This intuition can also corroborate the theoretical finding of regularization without bias term in Proposition 1, since the bias term corresponds to the vertical shift of the value function and has no impact on the Lipschitz constant of the value function.

---

> > ### Comment · Reviewer_r1LZ · 2023-08-13
> > **Response to rebuttal**
> >
> > The rebuttal addressed my concerns. I'll keep my rating.

---

> > > ### Author Response · Authors · 2023-08-16
> > >
> > > We are pleased that our rebuttal has addressed the reviewer's concerns. We greatly appreciate the reviewer's recognition of our work.

---

### Official Review · Reviewer_haxo · 2023-07-07

**Soundness:** 3 good
**Presentation:** 3 good
**Contribution:** 3 good
**Rating:** 6
**Confidence:** 2

**Summary:**

This paper studies the sim-to-real transfer problem. It extends the learning of a robust policy by using the framework of robust Markov decision processes (RMDPs). It extends this paradigm to large state and action spaces using two uncertainty set formulations: double sampling, and integral probability metric.
These formulations are then used in the proposed algorithm robust natural actor critic (RNAC). RNAC is tested in MuJoCo as well as on a real robot.

**Strengths:**

* Proposed uncertainty sets as well as the RNAC algorithm seem like practical steps forward in sim-to-real transfer with robustness approaches.
* The theoretical analysis seems interesting

**Weaknesses:**

* The experiments compare only to PPO. They should compare to other sim-to-real methods such as dynamics randomization [1] or action noise envelope [2].
* The related works being relegated to the appendix seems like a red flag. The authors should better organize the paper to include related works and comparisons to the main paper.

### References
[1] Peng, X.B., Andrychowicz, M., Zaremba, W. and Abbeel, P., 2018, May. Sim-to-real transfer of robotic control with dynamics randomization. In 2018 IEEE international conference on robotics and automation (ICRA) (pp. 3803-3810). IEEE.

[2] Jakobi, N., Husbands, P. and Harvey, I., 1995. Noise and the reality gap: The use of simulation in evolutionary robotics. In Advances in Artificial Life: Third European Conference on Artificial Life Granada, Spain, June 4–6, 1995 Proceedings 3 (pp. 704-720). Springer Berlin Heidelberg.


================================

The additional experiments alleviate my concerns about the experimental analysis.

**Questions:**

* In the second to last paragraph of Section 2, when describing the single worst-case kernel, is this kernel constant for the entire state space or varying?
* In the double sampling setup, does $\sum \alpha = 1$ in Equation (3)?

---

> ### Author Rebuttal · Authors · 2023-08-09
>
> Thank you very much for your comments and suggestions. We are encouraged by the fact that the reviewer finds that our paper takes "practical steps forward in sim-to-real transfer with robustness approaches" and has "interesting theoretical analysis". Please see our response below with respect to the specific comments. Please note that line numbers are based on the supplementary material (RNAC-full.pdf). We believe that we have addressed all the concerns, and we sincerely hope that the reviewer would consider increasing the score.
>
> **Q1.** "The experiments compare only to PPO. They should compare to other sim-to-real methods such as dynamics randomization or action noise envelope."
>
> **Response:** We compare the proposed algorithm with the reviewer's suggested baselines -- "dynamics randomization" and "action noise envelope" for MuJoco Envs and TurtleBot experiment.
>
> Please note that in Appendix A.3, we have in fact compared the proposed RNAC algorithms with soft actor-critic and soft-robust PPO algorithms in MuJoCo, and indeed reported results in Figure 5 with detailed explanation in Lines 557-582. The core idea of the soft robust algorithm is the same as "dynamics randomization", where the agent learns an optimal policy based on a distribution over an uncertainty set instead of considering the worst-case scenario. We add "action noise envelope" (Gaussian noise $\mathcal{N}(0, 0.05)$) as an additional baseline of Figure 5 in the attached pdf. In Figure 1 (attached new pdf), we also observe that the cumulative reward of RNAC-PPO decays much slower compared with action noise envelope, which further demonstrates the robust performance of the proposed algorithms.
>
> Additionally, based on the reviewer's suggestion, we have added two more baselines (i.e., "dynamics randomization" and "action noise envelope") to demonstrate the robustness of the proposed methods in the real-world TurtleBot environment. As shown in Figure 2 (attached new pdf), the proposed algorithms RNAC-PPO enjoy higher target-reaching rates (100\%) compared with action noise envelope (67.5\%), dynamic randomization (9\%), and PPO (0\%) under testing perturbed environments, which illustrates the robustness of the RNAC-PPO algorithms. Please see Lines 330-345 and Lines 584-600 for details of the TurtleBot environments. **Please also see Authors' Response to All.**
>
> **Q2.** "The related works being relegated to the appendix seems like a red flag. The authors should better organize the paper to include related works and comparisons to the main paper."
>
> **Response:**
> We have in fact discussed the most important related works quite extensively in the introduction. Please note that lines 30-66 are indeed a survey of related works, where almost all references have been mentioned. We leave the additional discussion of the technical sides of these works, e.g., convergence rate or sample complexity, to Appendix G, mostly due to space constraints. However, to avoid confusion, we will add a "related work" subsection under the introduction section in our revision.
>
> **Q3.** "In the second to last paragraph of Section 2, when describing the single worst-case kernel, is this kernel constant for the entire state space or varying?"
>
> **Response:** As adopted in all previous works in reinforcement learning literature, we refer to a map $p: \mathcal{S} \times \mathcal{A} \rightarrow \Delta_{\mathcal{S}}$ as a transition kernel. For each current state $s$ and current action $a$, the worst-case kernel $p(s'| s, a)$ is the probability that the next state is $s'$. It does depend on $s$.
>
> The second to the last paragraph of Section 2 summarizes the existing results on robust MDP in the literature [18, 37], so as to lay the foundation of robust RL. The major claim in the robust MDP with $(s,a)$-rectangular uncertainty set is that the optimal policy is stationary (not time-varying) and the worst-case transition for any stationary policy is also stationary.
>
> [18] Garud N Iyengar. Robust dynamic programming. Mathematics of Operations Research, 30(2):257–280, 2005.
>
> [37] Arnab Nilim and Laurent El Ghaoui. Robust control of markov decision processes with uncertain transition matrices. Operations Research, 53(5):780–798, 2005.
>
> **Q4.** "In the double sampling setup, does $\sum \alpha =1$ in Equation (3)?"
>
> **Response:** Yes, in Eq (3), $\sum_{i=1}^m \alpha_i = 1$ since $\alpha \in \Delta_{[m]}$, where $\Delta_{[m]}$ is a probability simplex, i.e., $\Delta_{[m]} = \{ \beta \in \mathbb{R}^m: \sum_{i=1}^m \beta_{i} = 1, \beta_i \geq 0, \forall~i\}$.

---

> > ### Comment · Reviewer_haxo · 2023-08-15
> > **Thank you for the response**
> >
> > I thank the authors for their response and for the additional experiments  taking into account the suggestions put forth in the review. These experiments now certainly showcase the effectiveness of the proposed technique better.
> >
> > I also thank the authors for answering the questions in the review.
> >
> > The related works could be better presented, but the authors' possible modifications might make the comparison sufficient.
> >
> > With these points, I will be revising my score.

---

> > > ### Author Response · Authors · 2023-08-17
> > >
> > > We are pleased that our rebuttal addresses the reviewer's concerns. We thank the reviewer for the valuable suggestions and recognition of our work. We would be glad to take action on any further suggestions from the reviewer to make our presentation better.

---

### Official Review · Reviewer_Fq66 · 2023-08-01

**Soundness:** 3 good
**Presentation:** 3 good
**Contribution:** 3 good
**Rating:** 6
**Confidence:** 1

**Summary:**

RL methods trained on simulators suffered from generalization problems because of the "simulation-to-reality-gap". Previous works proposed robust RL methods in a tabular setting, with limited search spaces. The paper aims to develop a computationally tractable robust RL algorithm with large search spaces. To this end, the paper proposed two novel uncertainty sets and the first policy-based approach for robust RL with provable convergence guarantees.

**Strengths:**

The paper studies a critical problem. Several technical contributions are proposed to devise a robust policy-based RL method with a large search space. The theoretical analysis seems solid.

**Weaknesses:**

The paper aims to devise a robust RL method. More real-world experiments are expected to demonstrate the robustness of the proposed methods.

---------------------
After reading the rebuttal, my main concerns were addressed.

**Questions:**

- Can previous tabulated-based robust RL methods be applied to the experiments in Sec. 7? If so, what about the comparison results? On the other hand, can the author provide some experiments to directly compare the proposed method with large search space to the previous one with limited search space to demonstrate the benefits of the proposed method?

**Limitations:**

The paper has discussed its limitations.

---

> ### Author Rebuttal · Authors · 2023-08-09
>
> We thank the reviewer for the comments and suggestions. We are encouraged by the fact that the reviewer finds that our paper "studies a critical problem" and provides "solid theoretical analysis". Please see our response below with respect to the specific comments. Please note that line numbers are based on the supplementary material (RNAC-full.pdf). We believe  that we have addressed all the concerns, and we sincerely hope that the reviewer would consider increasing the score.
>
> **Q1.** "More real-world experiments are expected to demonstrate the robustness of the proposed methods."
>
> **Response:**
> We would like to emphasize that our paper is primarily a   theoretical work that develops large-scale robust RL algorithms with provable convergence guarantees, the first of their kind to the best of our knowledge. We have also included extensive simulations of our RNAC algorithms on MuJoCo environments (Hopper-v3, Walker2d-v3, and HalfCheetach-v3) and have demonstrated their superior performance compared to many benchmarks. Significantly different from similar theoretical works on RL algorithms which limit the evaluations only to simulation experiments, we have gone one step further and demonstrated the effectiveness of our RNAC algorithm on a real-world mobile robot. **We have included the video of this real-world robot demonstration**, see Section 7.2 and the supplementary files.
>
> Now, based on your suggestion (also suggested by Reviewer haxo), we have added the results from additional MuJoCo and TurtleBot environments to illustrate the superior performance and effectiveness of our RNAC algorithm. **Please see Authors' Response to All for details**. We sincerely hope that these additional experiments will alleviate the reviewer's concerns.
>
> **Q2.** "Can previous tabulated-based robust RL methods be applied to the experiments in Sec. 7? Can the authors provide some experiments to directly compare the proposed method with large search space to the previous one with limited search space to demonstrate the benefits of the proposed method?"
>
> **Response:**
> We thank the reviewer for this suggestion. In Section 7, we run the proposed algorithm and non-robust baselines in large-scale MuJoCo (Hopper-v3, Walker2d-v3, and HalfCheetah-v3) and TurtleBot experiments with continuous state spaces and continuous action spaces. The uncertainty sets studied in previous works cannot scale up as discussed in Lines 42-49 and Appendix B, which makes them inapplicable to the experiments in Section 7. (Previous tabular-based papers show experimental results only in the tabular setting, as in [25, 57].) Therefore, new *large-scale* robust RL algorithms with corresponding convergence guarantees and superior empirical results are required, which is the major motivation of this work.
>
> [25] Navdeep Kumar, Esther Derman, Matthieu Geist, Kfir Levy, and Shie Mannor. Policy gradient for s-rectangular robust Markov decision processes. arXiv preprint arXiv:2301.13589, 2023.
>
> [57] Yue Wang and Shaofeng Zou. Policy gradient method for robust reinforcement learning. In International Conference on Machine Learning, pages 23484–23526. PMLR, 2022.

---

> > ### Comment · Reviewer_Fq66 · 2023-08-14
> >
> > The rebuttal has addressed my main concerns. I'll keep my rating.

---

> > > ### Author Response · Authors · 2023-08-17
> > >
> > > We thank the reviewer for the valuable comments on experimental evaluation and we are pleased to know that our rebuttal addresses the reviewer's concerns. Please let us know if you have any further questions. We will be happy to answer them. If you find our response satisfying, we wonder if you could kindly consider raising the score rating of our work?

---

### Author Rebuttal · Authors · 2023-08-09

## Authors' Response to All

We wholeheartedly thank all reviewers for their time and their constructive feedback on our paper. As suggested by Reviewers Fq66 and haxo, **we have added additional MuJoCo and real-world TurtleBot experiments** in the attached new pdf.

As suggested by Reviewer haxo, we have added more baselines (i.e., "dynamics randomization" and "action noise envelope") to demonstrate the robustness of the proposed methods in the real-world TurtleBot environment. As shown in Figure 2 (in the attached new pdf), the proposed algorithm RNAC-PPO enjoys higher target reaching rates (100\%), compared with action noise envelope (67.5\%), dynamic randomization (9\%), and PPO (0\%), under perturbed testing environments, which illustrates the robustness of the RNAC-PPO algorithm. Please see Lines 330-345 and Lines 584-600 for details of the TurtleBot environments.

We have also added one more baseline (i.e., "action noise envelope") to illustrate the robustness of the proposed methods in MuJoCo environments. In Figure 5 (original paper), we have already compared the proposed RNAC-PPO with PPO, soft actor-critic, and soft-robust PPO (same idea as dynamics randomization). Please see Lines 557-582 for a detailed description of the robust performance of RNAC-PPO. In Figure 1 in the attached new pdf, we also observe that the cumulative reward of RNAC-PPO decays much slower compared with action noise envelope, which further demonstrates the robust performance of the proposed algorithms.

*This paper closes an important gap in the policy-based approaches for robust RL under **function approximation** with **theoretical guarantee**.* Though a theory-driven paper with two novel uncertainty set designs and a robust policy-based algorithm with convergence guarantee, extensive experiments were conducted in the paper and more baselines are included during the rebuttal that lend credence to the theoretical contribution. We also hope that our detailed response will convince the reviewers of the value of our work and they will consider increasing their evaluations accordingly.

---

### Decision · Program_Chairs · 2023-09-21

**Decision:**

Accept (poster)

**Comment:**

The paper proposes an algorithm based on the natural actor critic that implements robustness, and presents both theoretical guarantees as well as experiments to show the advantage of the procedure.
Some limitations of the paper include limited numerical experiments and some organizational issues with respect to the appendix, and some limitations of the theoretical setup. But these seem to be addressable in a revised version as well as with further research.